# CalM: A Self-Supervised Foundation Model for Population Dynamics in Calcium Imaging Data

Xinhong Xu [*1]  Yimeng Zhang [*2]  Qichen Qian [2]  Yuanlong Zhang [2]

## Abstract

Recent work suggests that large-scale, multi-animal modeling can significantly improve neural recording analysis. However, for functional calcium traces, existing approaches remain task-specific, limiting transfer across common neuroscience objectives. To address this challenge, we propose **CalM**, a self-supervised neural foundation model trained solely on neuronal calcium traces and adaptable to multiple downstream tasks, including forecasting and decoding. Our key contribution is a pretraining framework, composed of a high-performance tokenizer mapping single-neuron traces into a shared discrete vocabulary, and a dual-axis autoregressive transformer modeling dependencies along both the neural and the temporal axis. We evaluate CalM on a large-scale, multi-animal, multi-session dataset. On the neural population dynamics forecasting task, CalM achieves competitive performance against strong specialized baselines after pretraining. With a task-specific head, CalM further adapts to the behavior decoding task and achieves superior results compared with supervised decoding models. Moreover, linear analyses of CalM representations reveal interpretable functional structures beyond predictive accuracy. Taken together, we propose a novel and effective self-supervised pretraining paradigm for foundation models based on calcium traces, paving the way for scalable pretraining and broad applications in functional neural analysis. Code is released at https://github.com/TSuXinH/CalM.

---
[*]Equal contribution  [1]Department of Automation, Tsinghua University, Beijing, China [2]School of Life Sciences, Tsinghua University, Beijing, China. Correspondence to: Yuanlong Zhang <yuanlongzhang@tsinghua.edu.cn>.

*Proceedings of the 43$^{rd}$ International Conference on Machine Learning*, Seoul, South Korea. PMLR 306, 2026. Copyright 2026 by the author(s).

## 1. Introduction

Modern systems neuroscience has witnessed a major research paradigm shift driven by advances in large-scale recording techniques such as Neuropixels (Jun et al., 2017; Steinmetz et al., 2021; Ye et al., 2025b) and functional calcium imaging (Barson et al., 2020; Sofroniew et al., 2016). It is now possible to record thousands of neurons simultaneously using either method and modern trace extraction pipelines (Stringer et al., 2019; Giovannucci et al., 2019), pushing the boundaries of neuroscience. While multi-session recordings across animals can provide new opportunities to better understand neural activities and behaviors, they create a pressing need for analysis frameworks that scale with dataset size and generalize beyond a single experiment or task (Paninski & Cunningham, 2018).

Recently, Transformers for neural data and neural foundation models (NFMs) have emerged as a promising approach to address this challenge (Azabou et al., 2023; 2024; Antoniades et al., 2023; Duan et al., 2025; Zhang et al., 2024; 2025b). NFMs are trained on multi-animal, multi-session datasets, and can generalize to held-out sessions, providing enhanced performance on tasks like encoding, decoding, and forecasting. Despite the progress, for functional calcium imaging data, existing approaches remain largely task-specific, focusing on behavior decoding (Azabou et al., 2023; 2024) or neural dynamics forecasting (Duan et al., 2025). This limits reuse and makes it difficult to establish a unified pretraining–fine-tuning paradigm for large-scale functional recordings.

To tackle this challenge, we propose CalM, a two-stage self-supervised autoregressive paradigm pretrained on large-scale calcium traces. Motivated by the intrinsic properties of calcium dynamics (Kerr & Denk, 2008) and the success of autoregressive generative models (Van Den Oord et al., 2017; Esser et al., 2021; Brown et al., 2020; Team et al., 2023; Tian et al., 2024; Singh et al., 2025), we first build a high-performance vector-quantized (VQ) tokenizer that converts single-neuron calcium traces into a shared discrete vocabulary. Then, a dual-axis autoregressive transformer modeling dependencies across the neurons within a population and across time within a trial is pretrained on top of the tokenized dataset. This design enables CalM to capture

structured population activity while maintaining compatibility with large multi-animal, multi-session datasets. After pretraining, CalM can be applied to the behavior task with a task-specific head.

We evaluate the model on simulated data and an open-source dataset recording mice performing a navigation decision task (Tseng et al., 2022), with 8 animals, 286 sessions and 273,770 neurons in total. We benchmark our framework on two representative tasks covering generation and understanding: neural population dynamics forecasting and behavior decoding. CalM achieves competitive performance on the forecasting task after self-supervised pretraining compared with specialized baselines. Moreover, by freezing the backbone and training the task-specific heads, CalM adapts to behavior decoding with superior results against models dedicated to decoding, suggesting that its representations capture both intrinsic neural dynamics and neurobehavioral relationships. Finally, we apply linear analysis to the model for interpretability, demonstrating that the neural embeddings exhibit clear functional distributions, and the forecasting results of the model effectively capture low-dimensional neural dynamics. Altogether, our work provides a general pretraining framework that supports diverse downstream tasks and provides meaningful biological insights.

The main contributions are summarized as follows:

- **Novel Tokenization.** We design a tokenization technique able to generate shared vocabulary for functional calcium imaging traces, facilitating downstream modeling.
- **Scalable Pretraining.** We introduce CalM, a self-supervised autoregressive pretraining framework for calcium traces, and scale it to a large multi-animal, multi-session dataset.
- **Versatile Applications.** We demonstrate that a single pretrained backbone supports forecasting and behavior decoding via task-specific heads, achieving competitive performance against strong specialized baselines.
- **Interpretability.** Through linear analysis, we show that CalM captures low-dimensional dynamics and provides emergent functional organization.

## 2. Related Work

### 2.1. Neural Population Dynamics Modeling

Modeling neural population dynamics is central in systems neuroscience. Many approaches posit low-dimensional latent state-space structure, from linear dynamical systems to switching variants that capture discrete regime changes (SLDS) and history-dependent transitions (rSLDS) (Fox et al., 2008; Linderman et al., 2016). In parallel, deep sequential latent-variable models based on RNNs, including LFADS-style approaches and related sequential autoencoders, infer single-trial latent trajectories (Sussillo et al.,

2016; Pandarinath et al., 2018). However, most of the methods are trained per dataset/session and do not directly handle heterogeneous neuron sets, limiting transfer across sessions and animals.

### 2.2. Neural–behavior Relationships and Decoding

Neural encoding/decoding spans classical GLM and reduced-rank formulations and modern deep learning approaches (Paninski, 2004; Truccolo et al., 2005; Paninski & Cunningham, 2018; Glaser et al., 2020). With increasing scale, work emphasizes cross-session / cross-subject robustness, including clinical BCI and representation learning that yields behavior-predictive embeddings tolerant to neuron and session variability (Gilja et al., 2015; Pandarinath et al., 2017; Willett et al., 2023; Metzger et al., 2023; Schneider et al., 2023). Yet many decoders remain task-specific and degrade under neuron turnover or session shifts, motivating more transferable representations.

### 2.3. Transformers for Neural Data

Recent large-scale recordings motivate foundation-model pretraining that transfers across tasks and sessions via self-supervised objectives and scalable transformers (Ye & Pandarinath, 2021; Ye et al., 2023; 2025a). Transformers have also been used to capture both individual-neuron and collective population structure (Liu et al., 2022). To handle heterogeneous neuron sets, POYO-style tokenization enables permutation-invariant multi-session learning, while multitask masking and scaling studies push toward broadly transferable representations (Azabou et al., 2023; 2024; Antoniades et al., 2023; Zhang et al., 2024). Multi-animal forecasting and reconstruction further highlight the need for neural-specific pretraining under session shifts (Duan et al., 2025; Xia et al., 2025). Representative generic MTS forecasters such as iTransformer (Liu et al., 2023) and PatchTST (Nie, 2022) ignore neuron-set variability. Overall, robustness to neuron turnover and broad downstream transfer remain open challenges.

Taken together, these works motivate neural foundation models that (1) scale to multi-animal, multi-session recordings and handle varying neuron sets, and (2) support diverse downstream objectives such as forecasting, behavior decoding, and representation analysis.

## 3. Methods

In this section, we describe the datasets and two-stage CalM framework in detail.

Our design is motivated by two complementary considerations. From a neuroscience perspective, calcium traces typically exhibit a fast rise followed by a slow, approximately exponential decay (Kerr & Denk, 2008). This stereo-

typed transient shape suggests that calcium signals can be effectively captured by a limited set of recurring patterns, making them well-suited for VQ. Provided that reconstruction quality is preserved, discrete VQ tokens can serve as an efficient surrogate for raw traces. From a deep learning perspective, modern large-scale generative models are predominantly trained to predict discrete tokens. In particular, cross-entropy (CE) loss optimizes the likelihood of the target token without requiring exact pointwise matching in continuous space, which is favorable for large-scale autoregressive training. In contrast, mean squared error (MSE) imposes strict constraints on amplitude deviations from the ground truth, which can be overly restrictive and less suitable for autoregressive modeling of long sequences.

### 3.1. Datasets

To comprehensively evaluate CalM, we leverage both simulated data and large-scale experimental data.

**Simulated data.** To assess the effectiveness of training and forecasting under controlled conditions, we simulate recurrent neural dynamics to generate multiple trials within a session using a fixed recurrent weight matrix $\mathbf{W}$ and stochastic initial states $\mathbf{r}_0$, governed by Eq. (1), where $\boldsymbol{\xi}$ denotes Gaussian noise.

$$\tau \frac{\mathrm{d}\mathbf{r}}{\mathrm{d}t} = -\mathbf{r} + \mathbf{W} \tanh(\mathbf{r}) + \boldsymbol{\xi} \tag{1}$$

**Collected data.** For experimental recordings, we utilize the open-source dataset introduced in (Tseng et al., 2022), where mice were trained to perform a dynamic navigation decision-making task. The full dataset comprises 8 subjects, 286 sessions, and 273,770 recorded neurons spanning 6 brain regions. For the neural forecasting task, we use trial-aligned neural activity. For the behavior decoding task, we use angular velocity signals along three rotational degrees of freedom (roll, pitch, and yaw).

To demonstrate the broad application of our model, we evaluate CalM on extra mouse (Sun et al., 2025) and *C. elegans* (Atanas et al., 2023) datasets. Further dataset details and preprocessing procedures are provided in Appendix A.

### 3.2. Neural Quantizer

We employ a Vector-Quantized Variational Autoencoder (VQ-VAE)-based architecture (Van Den Oord et al., 2017) as the neural quantizer (NQ). This module tokenizes continuous single-neuron calcium traces into discrete tokens, establishing a shared discrete vocabulary across the entire population (Figure 1A).

The NQ consists of an encoder $\mathcal{E}$, a decoder $\mathcal{D}$, and a codebook $\mathbf{E} \in \mathbb{R}^{K \times C}$, where $K$ is the codebook size and $C$ is the channel dimension. Given a trace $\mathbf{y} \in \mathbb{R}^T$,

the encoder first applies a convolutional layer to segment the trace into non-overlapping windows of length $L$, transforming them into temporal feature vectors $\mathbf{X}_w \in \mathbb{R}^{T_d \times C}$, where $T_d = \lfloor \frac{T}{L} \rfloor$. These vectors are then processed by Transformer layers to extract context-aware features $\mathbf{X} = \mathcal{E}(\mathbf{y}) \in \mathbb{R}^{T_d \times C}$, incorporating Rotary Positional Embeddings (RoPE) to encode sequence order (Su et al., 2024).

For each feature vector $\mathbf{x}_t \in \mathbb{R}^C$ at time step $t$, we identify the nearest codebook vector $\mathbf{e}_{i_t}$ based on Euclidean distance and pass the corresponding codebook entry to the decoder:

$$i_t = \arg \min_j \|\mathbf{x}_t - \mathbf{e}_j\|_2 \tag{2}$$

The decoder $\mathcal{D}$ comprises Transformer layers followed by a transposed convolutional layer, mapping the selected codebook vector sequence back to the reconstructed trace $\hat{\mathbf{y}}$:

$$\hat{\mathbf{y}} = \mathcal{D}\big([\mathbf{e}_{i_1}, \mathbf{e}_{i_2}, \ldots, \mathbf{e}_{i_{T_d}}]\big) \tag{3}$$

**Training objectives.** To ensure training stability, we initially freeze the codebook $\mathbf{E}$ and train the encoder $\mathcal{E}$ and decoder $\mathcal{D}$ for a warm-up period. We utilize MSE and Pearson correlation loss to ensure faithful reconstruction, balanced by a coefficient $\alpha$:

$$\mathcal{L}_{\mathrm{r}} = \mathrm{MSE}(\hat{\mathbf{y}}, \mathbf{y}) + \alpha\big(1 - \mathrm{Correlation}(\hat{\mathbf{y}}, \mathbf{y})\big) \tag{4}$$

To address the non-differentiability of the quantization operation in Eq. (2), we adopt the straight-through estimator (Bengio et al., 2013). Specifically, a commitment loss constrains $\mathbf{x}_t$ to stay close to the selected codebook vector, where $\mathrm{SG}[\cdot]$ denotes the stop-gradient operator:

$$\mathcal{L}_{\mathrm{c}} = \|\mathbf{x}_t - \mathrm{SG}[\mathbf{e}_{i_t}]\|_2^2 \tag{5}$$

**Codebook regularization.** To prevent index collapse, we encourage the codebook vectors to be diverse and uniformly utilized. First, we maximize the differential entropy of the codebook distribution, approximated via the Gumbel-Softmax operator $\mathrm{GS}_{\tau_H}[\cdot]$ with temperature $\tau_H$ (Jang et al., 2016). Additionally, we impose an orthogonality regularizer to enhance diversity among codebook vectors:

$$\mathcal{L}_{\mathrm{ent}} = -H\Big(\mathbb{E}_t\Big[\mathrm{GS}_{\tau_H}\big(-\|\mathbf{x}_t - \mathbf{e}_j\|_2^2\big)\Big]\Big) \tag{6}$$

$$\mathcal{L}_{\mathrm{orth}} = \big\|\mathbf{E}\mathbf{E}^T - \mathbf{I}\big\|_F^2 \tag{7}$$

Furthermore, we implement a "dead code" revival strategy. We maintain a registered queue buffer of recent temporal feature vectors. Codebook vectors with usage falling below a predefined threshold are periodically reset using random samples from this buffer.

**Autoregressive regularization.** To improve the temporal predictability of the discrete tokens and facilitate downstream autoregressive modeling, we attach an auxiliary head $\mathcal{A}$ to the encoder output. This head is trained to predict the

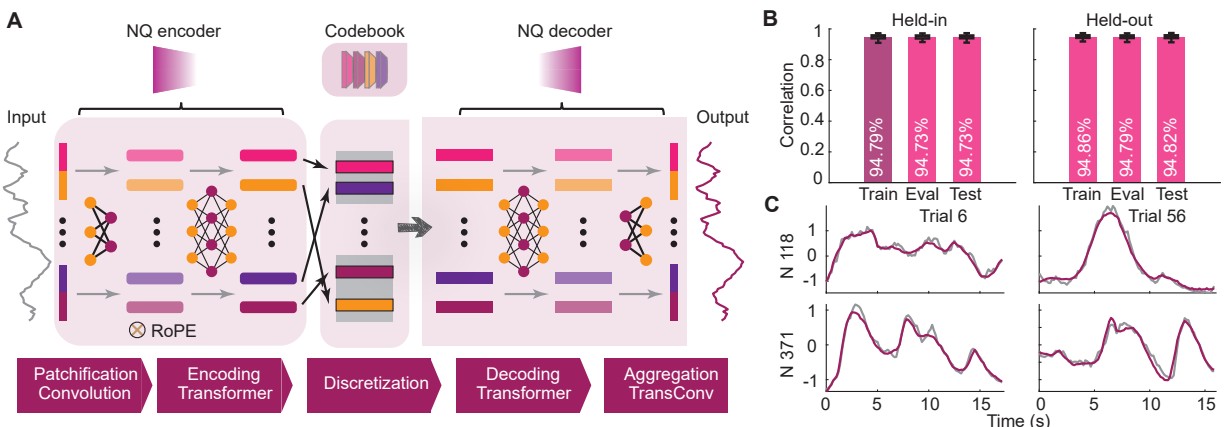

*Figure 1. NQ network and its performance.* (A) Details of NQ network. (B) Performance of NQ network on held-in and held-out datasets. We train the NQ network only on the training sets of the held-in datasets (burgundy), and apply the trained model to all the other datasets to do evaluation and generate tokenized datasets (pink). The numbers show the mean correlation for each bar. (C) Example neural traces from raw data (gray) and the NQ reconstruction results (burgundy).

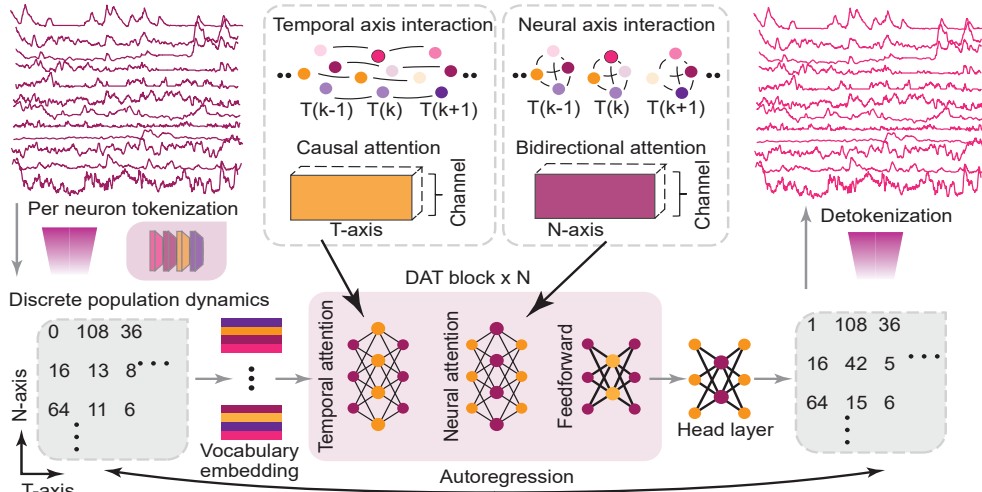

*Figure 2. DAT network and CalM framework.* We tokenize the traces and train the DAT model in an autoregressive manner with the tokens discretized with the NQ encoder. The predicted neural token matrix can be decoded into continuous trace space via NQ decoder.

next token, regularizing the latent space to capture temporal dependencies explicitly.

$$\mathbf{h}_t = \mathcal{A}(\mathbf{e}_{i_1}, \mathbf{e}_{i_2}, \ldots, \mathbf{e}_{i_t}) \in \mathbb{R}^C$$
$$p_\mathbf{A}(i_{t+1} = j \mid \mathbf{e}_{i_{1:t}}) = \mathrm{Softmax}(\mathbf{W}\mathbf{h}_t)_j \quad (8)$$
$$\mathcal{L}_{\mathrm{AR}} = \mathbb{E}_t\big[-\log p_\mathcal{A}(i_{t+1} \mid \mathbf{e}_{i_{1:t}})\big]$$

After applying these components, the NQ model can faithfully convert neural traces into discrete tokens for downstream pretraining. The total objective is formulated as:

$$\mathcal{L}_{\mathrm{total}} = \mathcal{L}_{\mathrm{r}} + \lambda_{\mathrm{c}}\mathcal{L}_{\mathrm{c}} + \lambda_{\mathrm{ent}}\mathcal{L}_{\mathrm{ent}} + \lambda_{\mathrm{orth}}\mathcal{L}_{\mathrm{orth}} + \lambda_{\mathrm{AR}}\mathcal{L}_{\mathrm{AR}} \quad (9)$$

### 3.3. Dual-Axis Transformer

With the trained NQ model, we tokenize trial-wise neural recordings into discrete sequences with compressed temporal resolution, denoted as $\mathbf{Z} \in \{1, 2, ..., K\}^{N \times T_d}$. This tokenized population activity is then fed into the Dual-Axis Transformer (DAT, $\mathcal{T}$), which serves as the neural foundation model for self-supervised learning (Figure 2).

**DAT backbone.** The DAT employs a Transformer architecture factorized along two axes to handle high-dimensional population data efficiently. Along the **neural axis**, DAT performs bidirectional self-attention across neurons within a single time step to capture population structure; along the **temporal axis**, it applies *causal* self-attention across time for each neuron to model temporal dynamics. We utilize RoPE to encode temporal positions.

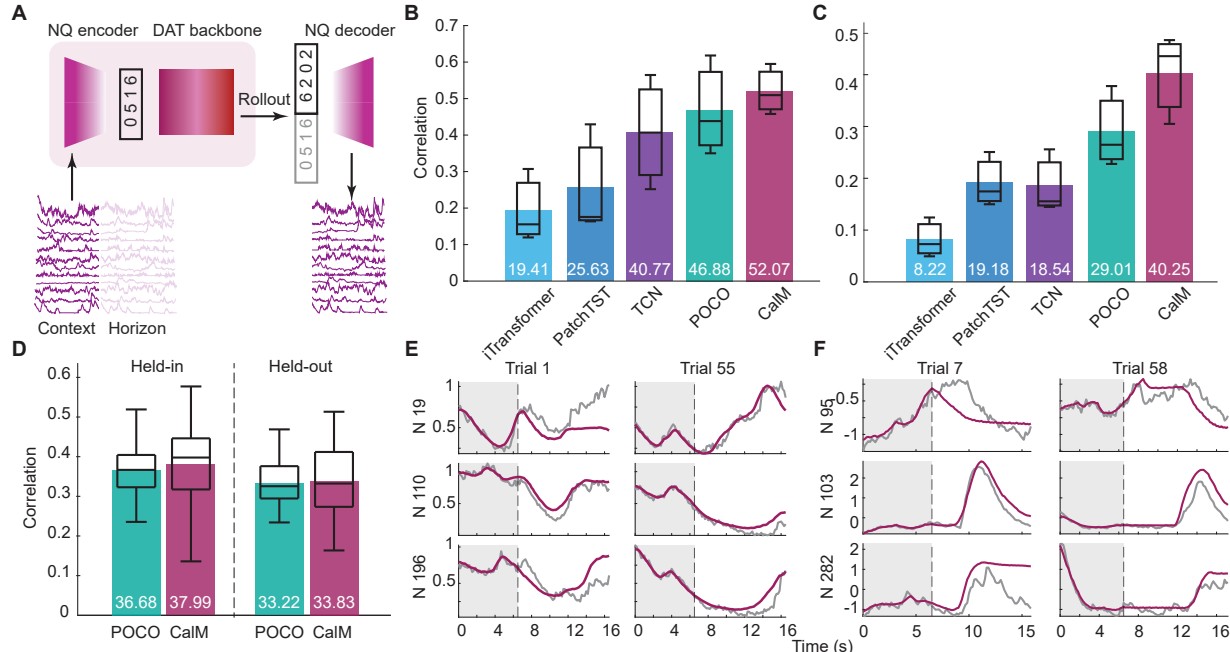

*Figure 3. Performance evaluation of CalM on neural population dynamics forecasting task.* (A) Illustration of how CalM implements forecasting task. (B, C) Performance of CalM against other baselines on simulated datasets (B) and single-session datasets (C). (D) Performance of CalM against POCO on multi-session datasets. (E, F) Example single neuron traces visualization of raw (gray) and CalM (burgundy) on simulated datasets (E) and collected datasets (F).

**Neuron and session embedding:** To preserve neuron identity, we assign a learnable embedding to each neuron. Furthermore, to account for cross-session variability in the multi-animal setting, we incorporate session embeddings to condition the model on specific recording contexts.

**Autoregressive objective.** Formally, given the history of tokens $\mathbf{Z}_{:,1:t}$, the model predicts the probability distribution for the next tokens $\mathbf{Z}_{:,t+1}$:

$$p(\mathbf{Z}_{:,t+1}|\mathbf{Z}_{:,1:t}) = \mathcal{T}(\mathbf{Z}_{:,1:t}) \qquad (10)$$

We optimize the model using the CE loss averaged over all neurons and time steps:

$$\mathcal{L}_{\text{CE}} = \mathbb{E}_t[-\log p_{\mathcal{T}}(\mathbf{Z}_{:,t+1} \mid \mathbf{Z}_{:,1:t})] \qquad (11)$$

**Auxiliary training strategies.** To improve training stability and rollout performance, we leverage two auxiliary losses:

*(i) Scheduled sampling.* To mitigate exposure bias, we randomly select a segment of length $s$ starting at $s_0$ and replace the input tokens with the model's one-step predictions (i.e., removing teacher forcing), yielding a perturbed sequence $\tilde{\mathbf{Z}}_{:,1:T_d}$. The model is then trained to predict the correct next tokens based on this perturbed history:

$$\mathcal{L}_{\text{SS}} = \text{CE}\Big(\mathcal{T}\big(\tilde{\mathbf{Z}}_{:,1:T_d-1}\big), \mathbf{Z}_{:,2:T_d}\Big) \qquad (12)$$

*(ii) Neighborhood replacement.* To enhance robustness against quantization errors, each input token $Z_{n,t}$ is replaced

with probability $p^{\text{nr}}$ to form a perturbed input $\mathbf{Z}^{\text{nr}}_{:,1:T_d-1}$ during training. Replacement tokens are sampled uniformly from the $k$ nearest neighbors, based on cosine similarity in the codebook space of the original token:

$$Z^{\text{nr}}_{n,t} = \begin{cases} Z_{n,t}, & \text{if } M_{n,t} = 0, \\ R_{n,t}, & \text{if } M_{n,t} = 1, \end{cases}$$

$$M_{n,t} \sim \text{Bernoulli}(p^{\text{nr}}), R_{n,t} \sim \text{Unif}(\text{Neigh}(Z_{n,t}))$$

$$\mathcal{L}_{\text{NR}} = \text{CE}\big(\mathcal{T}\big(\mathbf{Z}^{\text{nr}}_{:,1:T_d-1}\big), \mathbf{Z}_{:,2:T_d}\big) \qquad (13)$$

The final objective function for DAT is a weighted sum of the primary and auxiliary losses:

$$\mathcal{L}_{\text{total}} = \mathcal{L}_{\text{CE}} + \lambda_{\text{SS}}\mathcal{L}_{\text{SS}} + \lambda_{\text{NR}}\mathcal{L}_{\text{NR}} \qquad (14)$$

**Behavior decoding head.** Given latent features $\mathbf{h} \in \mathbb{R}^{N \times T_d \times C}$, the head outputs $M$ behavioral channels at the original temporal resolution $T = LT_d$. The head first computes $O = ML$ low-rate outputs, reshapes them to $\mathbb{R}^{M \times T}$, and applies a depthwise temporal convolution for smoothing. All heads are trained with an MSE objective. $\mathbf{B}_i$ and $\mathbf{b}$ denote learnable projection matrices and biases.

To assess the quality of the frozen representations, we first adopt a linear low-rank readout that aggregates neuron-wise information through learned weights.

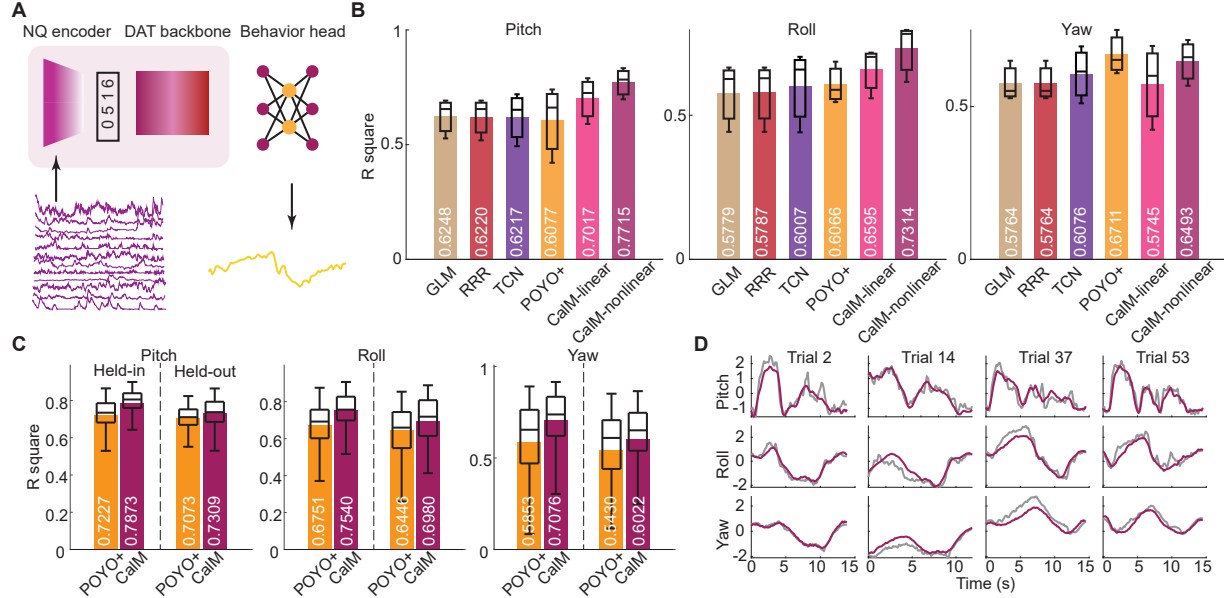

*Figure 4. Performance evaluation of CalM on behavior decoding.* (A) Schematic of the task-specific behavior decoding head. (B) Performance comparison of CalM with linear vs. nonlinear heads against baselines on three single-session datasets across three behavioral variables. (C) Performance comparison of CalM against POYO+ on multi-session datasets. (D) Representative visualization of decoding results for a single behavioral variable: ground truth (gray) and CalM prediction (burgundy).

$$y_{t,o} = \sum_{n=1}^{N} \sum_{r=1}^{R} \left( \sum_{c=1}^{C} h_{n,t,c} \left( \mathbf{B}_1 \right)_{o,r,c} \right) \left( \mathbf{B}_2 \right)_{o,n,r} + \left( \mathbf{b} \right)_o$$
$$\hat{\mathbf{u}} = \mathrm{DWConv} \left( \mathrm{Reshape}_{M,L}(\mathbf{y}) \right)$$
(15)

To further improve decoding performance, we employ a GLU-style (Shazeer, 2020) nonlinear head with dropout and GELU activation. Crucially, neuron-wise contributions are aggregated by querying a global neuron-weight bank using neuron indices $\mathrm{id}_n$ for varying neurons across sessions.

$$\mathbf{z}_{n,t} = \mathrm{Dr}\left( (\mathbf{h}_{n,t} \mathbf{B}_3) \odot \mathrm{GELU}(\mathbf{h}_{n,t} \mathbf{B}_4) \right)$$
$$y_{t,o} = \sum_{n=1}^{N} \sum_{r=1}^{R} z_{n,t,o,r} \left( \mathbf{B}_5 \right)_{o,\mathrm{id}_n,r} + \left( \mathbf{b} \right)_o \quad (16)$$
$$\hat{\mathbf{u}} = \mathrm{DWConv} \left( \mathrm{Reshape}_{M,L}(\mathbf{y}) \right)$$

## 4. Evaluation

### 4.1. Tasks

We evaluate CalM on two primary tasks, spanning both generative modeling and representation understanding:

**Neural population dynamics forecasting.** Given a context window of neural activity from the beginning of a trial, the model forecasts the population-level calcium traces for the remainder of the trial (forecast horizon).

**Behavior decoding.** Given the neural population activity of an entire trial, the model decodes continuous behavioral variables with the behavior head. Specifically, the discrete tokens representing the full trial are fed into the pretrained backbone to regress the corresponding velocity profiles.

### 4.2. Baselines

We benchmark CalM against a comprehensive set of competitive baselines in single-session experiments. For the more challenging multi-animal, multi-session settings, we compare against two state-of-the-art task-specific baselines: POCO for neural forecasting and POYO+ for behavior decoding. Details of baseline information and hyperparameter setting can be found in Appendix B.

## 5. Experiments

For all datasets, trials within each session are randomly partitioned into training (70%), validation (15%), and testing (15%) sets. In multi-session experiments, the entire dataset is split into held-in data (6 animals, 189 sessions, 197,704 neurons) and held-out data (2 animals, 97 sessions, 76,066 neurons) to establish a rigorous generalization benchmark. For the extra mouse dataset (Sun et al., 2025), held-in and held-out data contain 10 subjects with 30 sessions and 128,421 neurons, and 3 subjects with 9 sessions and 31,761 neurons, respectively. While for the *C. elegans* dataset (Atanas et al., 2023), held-in and held-out data contain 60 subjects with 60 sessions and 8,084 neurons, and 20 subjects

with 20 sessions and 2,698 neurons, respectively.

The multi-session NQ model is trained exclusively on the training sets of the held-in data and is subsequently evaluated on all datasets. For the DAT model, we train it on the tokenized held-in dataset. To evaluate generalization to held-out data, we freeze the pretrained backbone and optimize only the necessary session/neuron embeddings or task-specific heads. In single-session experiments, we randomly select three sessions from the held-out data. Simulation experiments are treated as single-session settings, where three groups of datasets are generated (governed by Eq. 1). Note that session embeddings are not used in single-session experiments. Details of the data, model and ablation studies can be found in Appendix A, C and D, respectively.

### 5.1. NQ Performance Evaluation

To assess the generalization capability of the shared-vocabulary tokenizer, we train the NQ model on the held-in training data and evaluate reconstruction quality on all held-in and held-out datasets. We use the resulting discrete tokens to generate the tokenized datasets required for DAT training.

### 5.2. Neural Population Dynamics Forecasting

For single-session settings, we compare CalM's forecasting performance against established baselines. For multi-session settings, we train the full model on the held-in dataset and fine-tune only the new session and neuron embeddings when adapting to held-out data.

We use a context window of 40 time points. The forecasting horizon begins immediately after the context and extends to the end of the trial. Importantly, we do not fine-tune the DAT backbone on raw traces during the forecasting stage. We perform evaluation via autoregressive rollout: we prefill the context tokens, predict the subsequent token sequence, and detokenize the output back to continuous traces using the NQ decoder. Regarding baseline comparison, POCO is designed with a fixed forecasting window setting to 24 time steps, since longer horizons would lead to substantial trial truncation, whereas CalM rolls out autoregressively to generate the full remainder of the trial. For the *C. elegans* dataset, the horizon is set to 40 time points.

### 5.3. Behavior Decoding

For single-session settings, we freeze the pretrained backbone, attach the task-specific behavior head, and fine-tune the head parameters. We then benchmark the model against behavior-related baselines. For multi-session settings, after pretraining and adapting CalM to held-out data to obtain complete session and neuron embeddings, we train the behavior head on held-in data. For adaptation to held-out

sessions, we fine-tune the head by optimizing only the learnable weights corresponding to the held-out neurons, keeping the backbone and the projection layers fixed.

### 5.4. Linear Analysis for Interpretability

Pretrained foundation models may yield rich representations that reveal underlying population dynamics. We analyze two key components of the trained DAT model to extract biological insights.

**Neural embeddings.** The learned neuron embeddings are expected to capture the functional identities of neurons. To validate this, we identify neurons in the held-in dataset that exhibit strong tuning to cue and choice variables quantified by $d'$ and pool them across sessions. We apply Principal Component Analysis (PCA) for dimensionality reduction and visualization (Jolliffe & Cadima, 2016). Furthermore, to explicitly examine how these embeddings encode task variables, Linear Discriminant Analysis (LDA) is leveraged to identify the axes that best discriminate between functional groups (Rao, 1948).

**Low-dimensional dynamics.** Despite pointwise forecasting errors, we investigate whether the low-dimensional dynamics of the predicted traces preserve the structure of the ground truth. As a proof of concept, we first apply PCA to the concatenated ground-truth traces over the horizon period to obtain a set of principal axes. We then project the forecasted traces of CalM and POCO onto these axes to extract latent trajectories. Finally, we compute the correlation between the predicted and ground-truth principal component trajectories. Shuffle tests by permuting trial identities within each session are made to assess statistical significance. This analysis quantifies how well the model-generated dynamics align with the intrinsic manifold of the neural population.

## 6. Results

### 6.1. NQ Performance Evaluation

The NQ model demonstrates high reconstruction quality, as well as strong generalization ability as it achieves similar performance on the other five evaluation datasets compared with the training dataset (Figure 1).

### 6.2. Neural Population Dynamics Forecasting

In simulated data and single-session experiments, CalM consistently outperforms strong baselines. In multi-session experiments, CalM remains competitive with POCO on both held-in and held-out datasets. Additionally, CalM performs comparably to POCO on the additional mouse dataset, but slightly worse on the *C. elegans* dataset (Table 1, 2). Notably, these results are achieved without directly optimizing CalM on raw traces or employing task-specific forecasting

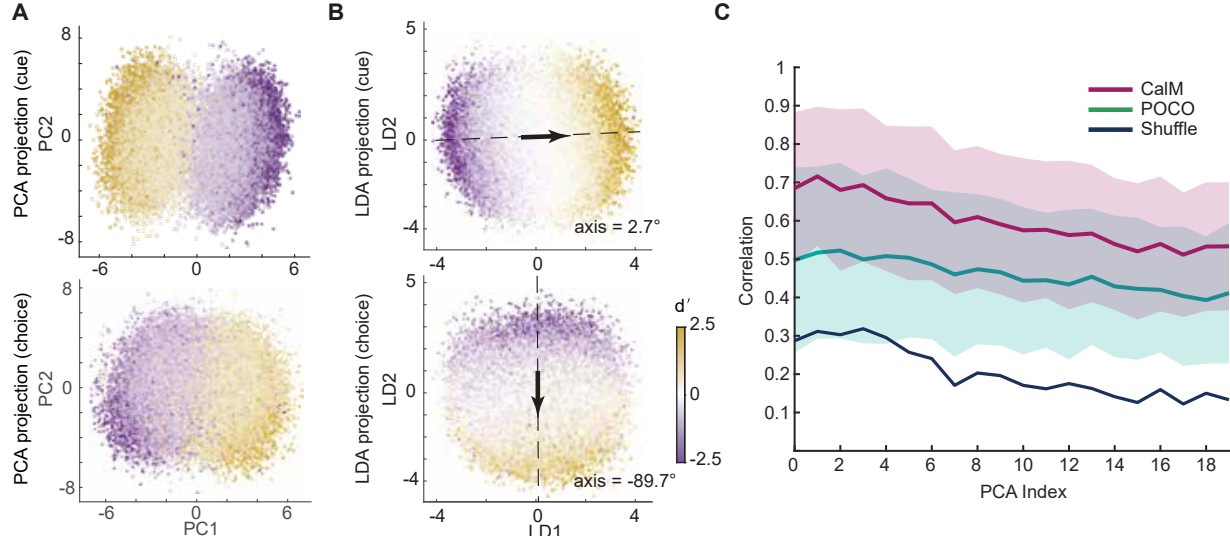

*Figure 5. Linear analysis for CalM framework.* (A) PCA visualization shows that neurons with strong tuning on cue or choice are well separated in an unsupervised manner. (B) LDA analysis of all the neural embeddings show that cue- and choice-encoding form orthogonal gradient structures. (C) Low dimensional dynamics of forecasting results from CalM correlate with ground truth more closely than POCO.

*Table 1.* Forecasting results for extra mouse dataset

| METRIC: CORR | HELD IN | HELD OUT |
|---|---|---|
| POCO | $0.2781\pm0.0343$ | $0.2568\pm0.0289$ |
| CALM | $0.2783\pm0.0562$ | $0.2592\pm0.0364$ |

*Table 2.* Forecasting results for *C. elegans* dataset

| METRIC: CORR | HELD IN | HELD OUT |
|---|---|---|
| POCO | $0.1935\pm0.0670$ | $0.1969\pm0.0468$ |
| CALM | $0.1841\pm0.0898$ | $0.1601\pm0.0757$ |

*Table 3.* Decoding position results for extra mouse dataset

| METRIC: $R^2$ | HELD IN | HELD OUT |
|---|---|---|
| POYO+ | $0.8810\pm0.0745$ | $0.8870\pm0.0649$ |
| CALM | $0.9706\pm0.0241$ | $0.9581\pm0.0376$ |

*Table 4.* Decoding velocity results for *C. elegans* dataset

| METRIC: $R^2$ | HELD IN | HELD OUT |
|---|---|---|
| POYO+ | $0.6922\pm0.2321$ | $0.7394\pm0.0950$ |
| CALM | $0.7065\pm0.2658$ | $0.7443\pm0.1010$ |

*Table 5.* Decoding head angle results for *C. elegans* dataset

| METRIC: $R^2$ | HELD IN | HELD OUT |
|---|---|---|
| POYO+ | $0.4597\pm0.1469$ | $0.4953\pm0.1145$ |
| CALM | $0.5149\pm0.1378$ | $0.5606\pm0.0807$ |

supervision. Furthermore, CalM handles variable forecasting horizons more flexibly than POCO, which is constrained by a fixed, predefined forecasting window (Figure 3).

## 6.3. Behavior Decoding

As shown in Figure 4, in single-session experiments, CalM with the proposed fine-tuned head outperforms all baselines, with the exception of yaw velocity where it slightly trails POYO+. Importantly, the competitive performance of CalM using a linear probe indicates that self-supervised pretraining captures not only intrinsic neural dynamics but also their functional relationship to behavioral variables.

In the multi-session setting, CalM with the fine-tuned head significantly outperforms pretrained POYO+ by 10.1% in average $R^2$ on held-in datasets, without updating the backbone weights or embeddings. For the most challenging variable, yaw, the improvement reaches up to 20.9%. On held-out datasets, CalM continues to outperform POYO+ with a 7.2% overall improvement, demonstrating robust gen-

eralization. Furthermore, CalM outperforms POYO+ on the extra datasets, as shown in Table 3, 4, 5.

The standard deviation values and statistical test of reported performance metrics can be found in Appendix E.

## 6.4. Linear Analysis for Interpretability

**Neural embeddings:** Through PCA projection, we observe that cue- and choice-related neurons naturally segregate into distinct clusters with clear boundaries (Figure 5A). Further LDA analysis on pooled embeddings across all animals and sessions reveals additional structure: although LDA does not enforce gradient effects or orthogonality, unlike the shuffled controls in Figure 7, we observe a continuous tuning-strength gradient and approximately orthogonal cue/choice

gradients (Figure 5B). This suggests that the four neuron groups encoding cue (black vs. white) and choice (left vs. right) are well separated in a linear latent space, indicating a functional organization under this task setting, which would be hard to recover from conventional single-session analyses alone. Complementary information can be found in Appendix F.

**Low-dimensional dynamics:** CalM exhibits higher correlation with ground-truth principal-component trajectories than POCO and substantially outperforms shuffled controls. Interestingly, these correlations exceed the average correlation observed for full population-trace forecasting. This indicates that, even when overall forecasting accuracy is comparable, CalM better captures the underlying low-dimensional dynamics (Figure 5C). This observation is reminiscent of autoregressive language modeling: while different valid token sequences can be generated from the same context, they preserve a consistent high-level structure within the representation space.

## 7. Discussion

In this work, we introduced CalM, a self-supervised neural foundation model pretrained on large-scale, multi-animal calcium imaging datasets. CalM effectively bridges **generative and discriminative tasks**, achieving competitive performance in both neural population forecasting and behavior decoding compared to current state-of-the-art methods. Beyond predictive accuracy, **linear analyses for interpretability** demonstrate that the learned representations exhibit clear functional segregation, and that the forecasted trajectories faithfully capture underlying low-dimensional population dynamics.

Despite these promising results, several avenues remain for future exploration. First, CalM currently relies on trial-aligned data, which may limit its broader applicability to spontaneous or continuously recorded activity. Trial alignment provides additional temporal structure and regularity; therefore, models trained under this setting may not directly transfer to less structured pseudo-trials without further adaptation. Second, although we intentionally avoid fine-tuning the backbone directly on traces for forecasting in the current framework, incorporating task-specific supervision or direct optimization in the trace space may further improve forecasting precision. Third, given CalM's strong performance in behavior decoding, the pretrained backbone may be extendable to a broader range of downstream regression and classification tasks through specialized task heads, which are currently not covered in this paper. Fourth, as modern neuroscience experiments increasingly combine neural activity with behavioral, anatomical, stimulus, and other contextual variables, extending CalM towards multimodal pretraining could improve its ability to integrate richer experimental information. Finally, since the current framework still consists of several modular components, an important direction is to streamline CalM into a more unified and end-to-end architecture that can generalize across heterogeneous datasets, which serves as a key property of general-purpose neural foundation models. More broadly, the emergent properties of CalM and its representation space warrant further investigation beyond the linear analyses presented in this study.

Looking forward, we hope this framework will inspire future research into large-scale pretraining for neural data. By providing a scalable approach to learning from multi-animal datasets, CalM moves us closer to neural foundation models that not only excel in predictive performance but also offer generalizable insights to accelerate neural discovery.

## Impact Statement

This work proposes a self-supervised foundation model for calcium imaging population activity to learn transferable neural representations and improve downstream analyses such as decoding and forecasting. It may reduce task-specific engineering and support more data-efficient neuroscience research.

Potential risks include misuse or over-interpretation of learned representations, and privacy concerns if similar approaches are applied to human neural data. This paper makes no clinical claims and focuses on public non-human animal datasets; any extension to human recordings should require ethical review, informed consent, strong de-identification and access control, and careful reporting of limitations and uncertainty.

## Acknowledgement

This work is supported by the National Natural Science Foundation of China (62522510) and Tsinghua Dushi Program.

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

# A. Dataset

## Collected dataset

**Tseng dataset:** We use the open-source dataset from (Tseng et al., 2022). The full dataset comprises calcium recordings from 8 mice across 286 sessions, totaling 273,770 neurons recorded from 6 brain regions and 2 cortical layers during a rule-switching decision-making task, with an average of approximately 957 neurons per session. In this task, mice are placed in a Y-maze, where they receive sustained black or white visual stimuli (cues) in the stem section. In the subsequent arm section, each mouse is required to perform a corresponding left or right turn. The cue–response rule is switched every 100 to 150 epochs. The original sampling rate is 30 Hz. Neurons from different imaging planes within a single scan are considered to be recorded simultaneously; therefore, we use the effective sampling rate of 6 Hz. The dataset was partitioned into held-in (6 subjects, 189 sessions, 197,704 neurons) and held-out (subject 6 and 10, 97 sessions, 76,066 neurons) sessions by mouse subjects. Within each session, data are further segmented into trials following the provided dataset annotations. For each session, trials are randomly assigned to the training, validation, and test sets in a 70:15:15 ratio. The details of the data distribution are summarized below.

*Table 6.* Dataset distribution on mouse subject

| SUBJECT ID | 3 | 4 | 5 | 6 | 7 | 8 | 9 | 10 |
|---|---|---|---|---|---|---|---|---|
| NUMBER OF SESSIONS | 17 | 23 | 31 | 46 | 35 | 42 | 41 | 51 |
| NUMBER OF NEURONS | 17861 | 20592 | 34293 | 40262 | 36621 | 42550 | 45787 | 35804 |
| MEAN TRIAL FRAMES | 87 | 105 | 90 | 83 | 90 | 95 | 77 | 104 |

*Table 7.* Dataset distribution on brain region

| REGION | AM | MM | PM | RSC | V1 | VISA |
|---|---|---|---|---|---|---|
| NUMBER OF SESSIONS | 42 | 41 | 35 | 55 | 43 | 70 |
| NUMBER OF NEURONS | 42088 | 44520 | 32291 | 51262 | 39619 | 63990 |
| MEAN TRIAL LENGTH | 90 | 93 | 91 | 90 | 92 | 92 |

**Sun dataset:** For the mouse dataset from (Sun et al., 2025), which involves a cue–delay–choice navigation task in which mice must use a transient visual cue to infer trial type and select the correct future reward location, we select a subset sampled at 10 Hz, which contains 13 subjects and a total of 160,182 neurons across 39 sessions. We use the first 64 time points of each trial for forecasting and decoding.

**Atanas dataset:** For the *C. elegans* dataset from (Atanas et al., 2023), which consists of brain-wide calcium imaging of spontaneous activity in freely moving animals, we use 80 released HDF5 recording datasets after excluding scrambled-control recordings. Each HDF5 file corresponds to an individual per-animal recording dataset and was treated as one session in our preprocessing and modeling pipeline. Therefore, in our dataset statistics, we report the Atanas dataset as containing 80 animals / sessions and 10,782 neurons recorded at approximately 1.7 Hz, with an average of 135 head neurons per session. In our processed subset, we split the data into 60 held-in and 20 held-out sessions, which are segmented into non-overlapping windows of 80 time points as pseudo-trials.

For all collected datasets, we preprocess the neural signals by first applying an exponential moving average with $\alpha = 0.15$ for temporal smoothing of calcium traces within sessions. The smoothed traces are then segmented into trials, which are randomly split into training, validation, and test sets. Finally, we compute the per-neuron mean and standard deviation using only the training split and use these statistics to z-score normalize the neural data in all three splits.

## Simulation dataset

For the simulation dataset, we use a calcium signal simulation pipeline based on a recurrent neural network (RNN). Hidden states in RNN follow the dynamics in Eq. 1 with stochastic initial $\mathbf{r}_{0,i} \sim \mathcal{N}(0, 0.01)$ and process noise $\xi_i \sim \mathcal{N}(0, 0.01)$. The weights $\mathbf{W}_{ij}$, which are kept fixed across all trials within a session, are i.i.d. sampled from a Gaussian distribution $\mathcal{N}(0, \frac{g^2}{N})$ where gain $g = 1.6$ and $N$ is the number of neurons.

To simulate calcium signals, we generate spike trains $\mathbf{S}_t$ from the post-activation activity of the RNN using a Poisson

process, as shown in Eq. 17, with time step $dt = 0.2$ s and maximum spike rate $\lambda_{\max} = 5.0$. The spike trains are then convolved with a bi-exponential calcium dynamics kernel $\mathcal{K}$ in Eq. 18 with rise time constant $\tau_r = 0.05$ and decay time constant $\tau_{\mathrm{ca}} = 2.0$ to produce simulated calcium traces. Finally, small observational noise $\epsilon_i \sim \mathcal{N}(0, 0.016^2)$ is added to the traces.

$$\mathbf{S}_t \sim \mathrm{Poisson}(\frac{\tanh(\mathbf{r}_t) + 1}{2} \times dt \times \lambda_{\max}) \tag{17}$$

$$\mathcal{K} = \exp\left(-\frac{t}{\tau_r}\right) - \exp\left(-\frac{t}{\tau_{\mathrm{ca}}}\right) \tag{18}$$

We generate three sessions using three different random seeds. Each session consists of 400 trials, which are split into training, validation, and test sets with a ratio of 70:15:15. Each trial contains calcium traces from 200 neurons over 100 time steps with a time step of 0.2 s.

## B. Baselines and Training Details

The following baselines follow the same data splits and per-trial training protocol as described above. For the forecasting task, we fix the context length to 40 time steps and use the Pearson correlation coefficient as the evaluation metric. For the decoding task, model checkpoints and best hyperparameters are selected based on validation MSE, and performance is reported using both the coefficient of determination ($R^2$) and MSE.

### General Linear Model (GLM)

For GLM (Paninski, 2004; Truccolo et al., 2005), neural activity is augmented using causal temporal lags for 4 steps ensuring that no future information leaks. Ridge regression with the regularization coefficient $\alpha$ selected from a logarithmic grid $[10^{-3}, 10^2]$ based on validation MSE is applied to decode behavior.

### Reduced Rank Regression (RRR)

For RRR (Izenman, 1975; Zhang et al., 2025a), we reuse the same lagged design and regularization strategy in GLM and the rank is chosen from $1, ..., d_y$ where $d_y$ denotes the dimension of behavioral output, based on validation MSE.

### Temporal Convolutional Network (TCN)

For the decoding task, we use a residual TCN (Bai, 2018) consisting of 4 causal convolutional blocks with a kernel size of 5 and a hidden dimension of 64. Each block uses two weight-normalized one-dimensional convolutional layers, ReLU activations, dropout, and a residual connection. A dropout rate of 0.2 is applied after each convolutional layer. The model is trained with MSE loss for up to 200 epochs using the Adam optimizer with learning rate $10^{-3}$ and batch size 32. Weight decay is selected from $[0, 10^{-5}, 10^{-4}, 10^{-3}]$ based on validation MSE, and early stopping with a patience of 20 epochs is applied.

For the forecasting task, we use TCN consisting of 5 causal convolutional layers with a kernel size of 5 and a hidden dimension of 256. A dropout rate of 0.1 is applied. During evaluation, the model performs autoregressive forecasting by iteratively predicting the next time step and appending it to the input sequence. The model is trained for 200 epochs using the Adam optimizer with a learning rate of $10^{-4}$ and a batch size of 8.

### PatchTST

PatchTST (Nie, 2022) is implemented using the NeuralForecast (Olivares et al., 2022). Neural activity is represented as a collection of univariate time series. PatchTST tokenizes the input sequence into overlapping temporal patches. In our implementation, the patch length is set to 8 time steps with a stride of 4. Each patch is projected into a latent space of dimension 128. The model consists of 3 transformer layers with 8 attention heads per layer and uses a dropout rate of 0.1. The model is trained using the Adam optimizer with a learning rate of $10^{-3}$ and is evaluated in a similar autoregressive manner.

### iTransformer

iTransformer (Liu et al., 2023) is also implemented using the NeuralForecast framework and follows the same data representation and evaluation protocol as PatchTST. We set the model dimension 64 with 4 attention heads and 2 transformer

layers. Dropout is set to 0.1. The model is trained using the Adam optimizer with a learning rate of $10^{-3}$.

**POCO**

For POCO (Duan et al., 2025), we preprocess all real datasets by truncating each trial to a fixed length of 64 time steps (40 for context and 24 for forecasting). Trials shorter than this length were discarded in accordance with the POCO design. We set the sliding window stride to 64 and the patch length sufficiently large to ensure that no trial is truncated or concatenated with another during training or evaluation. For the simulation dataset, we similarly set sliding window stride to 100, with 40 time steps as context and 60 for forecasting. We perform a broad hyperparameter search and choose the best based on validation MSE and Pearson correlation coefficient. Detailed hyperparameter settings are provided in Table 8. Due to the parameter stability of POCO, we use a single set of model parameters across all tasks, adjusting only the learning rate and weight decay. Only for the simulation dataset we use a compression factor of 10 to match the trial length.

*Table 8.* Hyperparameters used for training POCO model

| HYPERPARAMETER | VALUE |
| --- | --- |
| HIDDEN SIZE | 128 |
| LAYERS | 1 |
| NUMBER OF HEADS | 16 |
| LATENT TOKENS | 8 |
| FFN HIDDEN SIZE | 1024 |
| COMPRESSION FACTOR | 8 |
| BATCH SIZE | 64 |

**POYO+**

For POYO+ (Azabou et al., 2024) on multi-session decoding task, we perform a broad hyperparameter search on a small dataset containing 9 sessions and apply the best hyperparameters to the full 189 pre-train dataset. Only the learning rate is adjusted to ensure effective training. For single session decoding, we perform a grid search on model size, latent step, number of latents and dropout ratio, then choose the best hyperparameters based on validation MSE.

*Table 9.* Hyperparameters used for training POYO+ model

| HYPERPARAMETER | MULTI SESSION | SINGLE SESSION |
| --- | --- | --- |
| EMBEDDING DIMENSION | 128 | 64 |
| HEAD DIMENSION | 64 | 64 |
| NUMBER OF LATENTS | 32 | 32 |
| LATENT STEP | 0.125 | 0.1 |
| DEPTH | 4 | 4 |
| NUMBER OF HEADS | 8 | 8 |
| FFN DROPOUT | 0.2 | 0.0 |
| LINEAR DROPOUT | 0.4 | 0.0 |
| ATTENTION DROPOUT | 0.2 | 0.0 |
| BATCH SIZE | 64 | 64 |

# C. Model and Hyperparameters

Hyperparameters for training multi-session CalM are as follows. The NQ encoder and DAT are made into a causal system along the temporal axis during pretraining stage to prevent information leakage.

*Table 10.* Hyperparameters used for training CalM (NQ) model

| HYPERPARAMETER | VALUE |
|---|---|
| EMBEDDING DIMENSION | 512 |
| CODEBOOK SIZE | 128 |
| ENCODER / DECODER LAYERS | 4 |
| ATTENTION HEADS | 4 |
| DISCRETIZATION WINDOW / OVERLAP | 4 |
| EMA DECAY | 0.99 |
| GUMBEL TEMPERATURE (START $\rightarrow$ END) | $2.5 \rightarrow 0.01$ |
| ENCODER CAUSAL | TRUE |
| MAX AR HORIZON | 4 |
| $w_{embed}, w_{commit}$ | $1.0, 0.5$ |
| $w_{entropy}$ | $0.5$ |
| $w_{AR-CE}, w_{AR-Align}$ | $0.5, 0.5$ |

*Table 11.* Hyperparameters used for training CalM (DAT) model

| HYPERPARAMETER | VALUE |
|---|---|
| MODEL DIMENSION | 512 |
| LAYERS | 6 |
| ATTENTION HEADS | 8 |
| FFN DIMENSION | 2048 |
| SCHEDULED SAMPLING PROBABILITY | 0.6 |
| SCHEDULED SAMPLING BLOCK LENGTH | 6 |
| NEIGHBORHOOD REPLACEMENT PROBABILITY | 0.1 |
| NUMBER OF NEIGHBORS | 12 |

## D. Ablation Study

To evaluate the effect of data preprocessing on model performance, we vary the EMA smoothing coefficient in the single-session setting. Table 12 shows preprocessing mainly affects reconstruction/forecasting correlation, while decoding remains stable. We also observe the same qualitative trend: from $\alpha = 0.15$ to $1.00$, forecasting correlation performance drops by 34.7% for CalM and 39.3% for POCO, suggesting that this sensitivity is not unique to CalM, while CalM still performs better.

*Table 12.* Ablation results for different alpha values

| ALPHA | NQ RECON | DAT FORECAST | POCO FORECAST | MEAN DECODING $R^2$ |
|---|---|---|---|---|
| 0.15 (DEFAULT) | $0.9561 \pm 0.0114$ | $0.4600 \pm 0.0767$ | $0.3328 \pm 0.0706$ | $0.7499 \pm 0.0138$ |
| 0.60 | $0.8984 \pm 0.0100$ | $0.3603 \pm 0.0727$ | $0.2470 \pm 0.0501$ | $0.7746 \pm 0.0192$ |
| 1.00 (W/O SMOOTHING) | $0.8784 \pm 0.0213$ | $0.3003 \pm 0.0769$ | $0.2020 \pm 0.0450$ | $0.7796 \pm 0.0258$ |

To evaluate the necessity and effectiveness of our training strategies, we perform an ablation study on a small dataset consisting of 12 randomly selected sessions, and evaluate the forecasting performance of the pretrained DAT model using the correlation coefficient as the metric. Specifically, for the NQ, we test models without codebook regularization (the Gumbel-Softmax operator or the orthogonality regularizer) or without autoregressive regularization. For the DAT, we evaluate the effect of disabling scheduled sampling or neighborhood replacement during training. The ablation results, summarized in Table 13 and Table 14, show that all these training strategies contribute to improved CalM performance. For NQ, autoregressive regularization is the main driver of downstream forecasting gains, likely by encouraging temporally predictive tokens. Entropy and orthogonality regularization further help by preventing codebook collapse. For DAT, scheduled sampling is the most important component, as it mitigates exposure bias in long-horizon rollout, while neighborhood replacement provides a smaller gain by improving robustness to quantization errors.

*Table 13.* Ablation results for NQ

| METRIC | BASELINE | W/O AR LOSS | W/O GUMBEL | W/O ORTHOGONAL LOSS |
|--------|----------|-------------|------------|---------------------|
| AR CORR | 0.3424±0.0692 | 0.2968±0.1098 | 0.3148±0.0826 | 0.3124±0.1194 |

*Table 14.* Ablation results for DAT

| METRIC | BASELINE | W/O SCHEDULED SAMPLING | W/O NEIGHBORHOOD REPLACEMENT |
|--------|----------|------------------------|------------------------------|
| AR CORR | 0.3058±0.1098 | 0.2079±0.1027 | 0.2870±0.1160 |

## E. Numerical Results

Here we report the standard deviations ($\sigma$) of the results presented in the main text, as well as statistical tests for the multi-session experimental results. Wilcoxon tests show that CalM significantly outperforms POYO+ on decoding and POCO on held-in forecasting, while the held-out forecasting difference is not statistically significant.

*Table 15.* Reconstruction result with $\sigma$

| METRIC (CORR) | HELD-IN | HELD-OUT |
|---------------|---------|----------|
| TRAIN | 0.9479±0.0142 | 0.9486±0.0147 |
| EVAL | 0.9473±0.0143 | 0.9479±0.0149 |
| TEST | 0.9473±0.0142 | 0.9482±0.0147 |

*Table 16.* Single-session forecasting results with $\sigma$

| METRIC (CORR) | REAL | SIMULATION |
|---------------|------|------------|
| ITRANSFORMER | 0.0822±0.0383 | 0.1941±0.0996 |
| PATCHTST | 0.1918±0.0526 | 0.2563±0.1501 |
| TCN | 0.1854±0.0613 | 0.4077±0.1564 |
| POCO | 0.2901±0.0783 | 0.4688±0.1365 |
| OURS | 0.4025±0.0858 | 0.5207±0.0692 |

*Table 17.* Single-session decoding results with $\sigma$

| METRIC ($R^2$) | PITCH | ROLL | YAW |
|----------------|-------|------|-----|
| GLM | 0.6248±0.0864 | 0.5779±0.1190 | 0.5764±0.0650 |
| RRR | 0.6220±0.0912 | 0.5787±0.1195 | 0.5764±0.0649 |
| TCN | 0.6217±0.0948 | 0.6007±0.1145 | 0.6076±0.0762 |
| POYO+ | 0.6077±0.1664 | 0.6066±0.0719 | 0.6711±0.0725 |
| OURS-LINEAR | 0.7017±0.1017 | 0.6595±0.0875 | 0.5745±0.1391 |
| OURS-NONLINEAR | 0.7715±0.0687 | 0.7314±0.1003 | 0.6493±0.0755 |

*Table 18.* Statistical test for multi-session forecasting

| METRIC (CORR) | HELD-IN | HELD-OUT |
|---------------|---------|----------|
| POCO | 0.3668±0.0605 | 0.3322±0.0511 |
| OURS | 0.3799±0.0978 | 0.3383±0.0860 |
| DIFF | +0.0131 | +0.0061 |
| P-VALUE | $7.82 \times 10^{-4}$ | 0.3684 |
| SIGNIFICANCE | *** | N.S. |

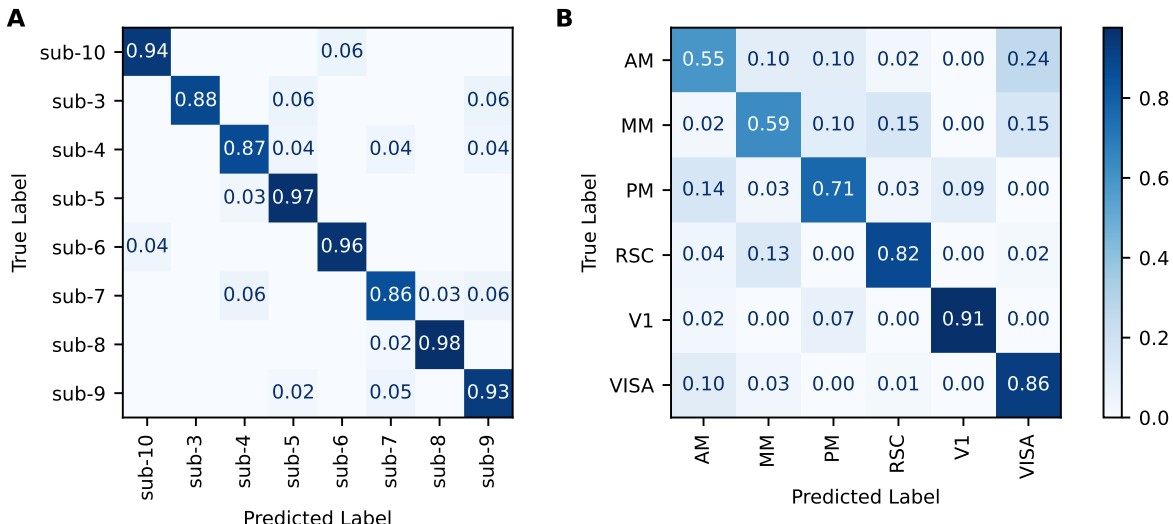

*Figure 6. Confusion matrices for classification using CalM session embedding.* (A) Mouse subject classification. (B) Brain region classification.

*Table 19.* Statistical test for multi-session decoding

| METRIC ($R^2$) | HELD-IN PITCH | HELD-IN ROLL | HELD-IN YAW | HELD-OUT PITCH | HELD-OUT ROLL | HELD-OUT YAW |
|---|---|---|---|---|---|---|
| POYO+ | 0.7227±0.0840 | 0.6751±0.1132 | 0.5853±0.2321 | 0.7073±0.0611 | 0.6446±0.1299 | 0.5430±0.2240 |
| OURS | 0.7873±0.0737 | 0.7540±0.0883 | 0.7076±0.1691 | 0.7309±0.0785 | 0.6980±0.1200 | 0.6022±0.2172 |
| DIFF | +0.0647 | +0.0790 | +0.1223 | +0.0236 | +0.0534 | +0.0592 |
| P-VALUE | $2.09 \times 10^{-32}$ | $1.47 \times 10^{-32}$ | $7.37 \times 10^{-30}$ | $2.59 \times 10^{-6}$ | $1.84 \times 10^{-10}$ | $2.14 \times 10^{-7}$ |
| SIGNIFICANCE | *** | *** | *** | *** | *** | *** |

## F. Representation Study

The trained CalM neural and session embeddings may form meaningful representations related to structural or functional identity. To explore the information encoded in these representations, we conducted a linear analysis. First, we evaluate whether the embeddings encode identity-related information by training a logistic regression classifier with a 5-fold cross-validation on the pooled embeddings, including both held-in and held-out samples. We evaluate the classification performance for both neural embeddings and session embeddings on three attributes: brain region, cortical layer (layer 2/3 vs. layer 5), and mouse identity. As shown in Table 20 and Figure 6, session embeddings exhibit high performance for layer and mouse identity, and achieve relatively good performance on brain region classification. In contrast, neural embeddings show substantially weaker performance across all attributes.

*Table 20.* Classification accuracies for neural and session embedding.

| EMBEDDINGS | SUBJECT | BRAIN REGION | LAYER |
|---|---|---|---|
| NEURAL | 0.224±0.001 | 0.184±0.001 | 0.621±0.002 |
| SESSION | 0.930±0.025 | 0.755±0.025 | 1.000±0.000 |

We further analyze whether the learned neuron embeddings represent task-related neuronal functional identities. For each held-in and held-out dataset, we quantified single-neuron tuning to task-related variables, cue and choice, using the statistic $d'$, defined in Eq. 19, where $\mu_1$ and $\mu_2$ denote the mean neuronal activity under two conditions, and $\sigma_1^2$ and $\sigma_2^2$ denote the corresponding variances across trials. It measures differences in neuronal activity across trials. Using this metric, we computed the tuning strength of each neuron with respect to cue and choice conditions.

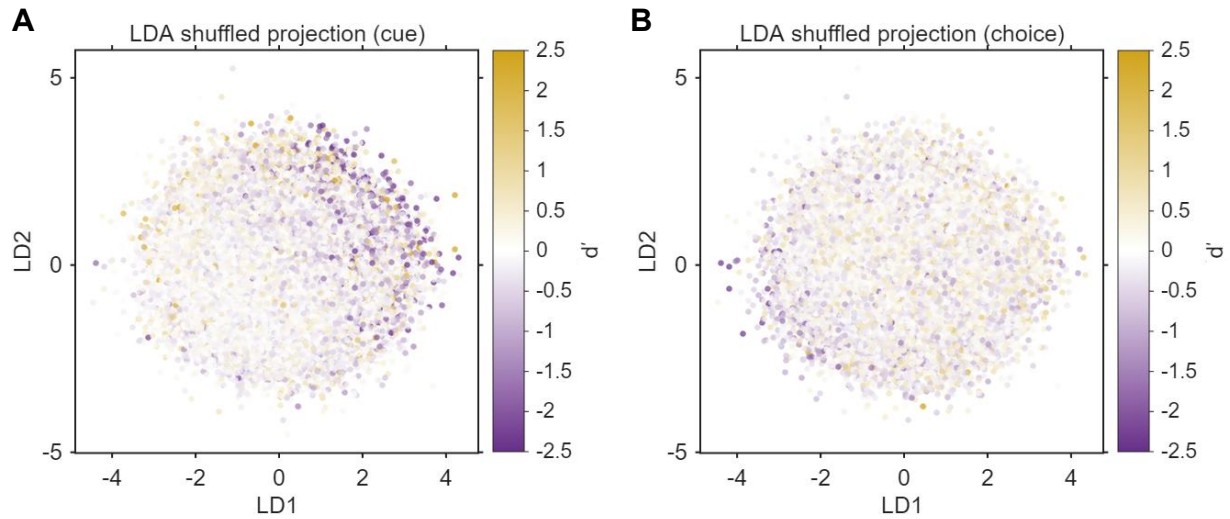

*Figure 7. Shuffle analysis of the LDA structure shown in Figure 5.* (A) Shuffle for LDA structure of cue-encoding for all held-in neural embeddings. (B) Shuffle for LDA structure of choice-encoding for all held-in neural embeddings.

$$d' = \frac{\mu_1 - \mu_2}{\sqrt{\frac{1}{2}(\sigma_1^2 + \sigma_2^2)}} \tag{19}$$

For cue-related tuning, we focused on a time window defined in the dataset within each trial, corresponding to the presentation of the visual stimulus (stem section). For choice-related tuning, we considered a later time window, during which the mouse performs a choice (arm section). For strongly tuned neurons, defined as $|d'| > 0.5$, PCA of neural embeddings reveals clear clustering structure as shown in Figure 5A and 8A. We also performed a four-way LDA on neural embeddings corresponding to the four combinations of cue and choice conditions (positive and negative for each variable). LDA projections exhibit a clear gradient structure in the LDA plane. We defined the principal axis for each task variable as the direction in the LDA plane that maximizes the correlation coefficient between coordinates along the axis and the corresponding tuning strength. In both the held-in and held-out datasets, the resulting correlations are significantly higher than those obtained from shuffled controls, and the principal axes associated with cue and choice tuning are approximately orthogonal, as shown in Figure 5B, Figure 7 and Figure 8B,C.

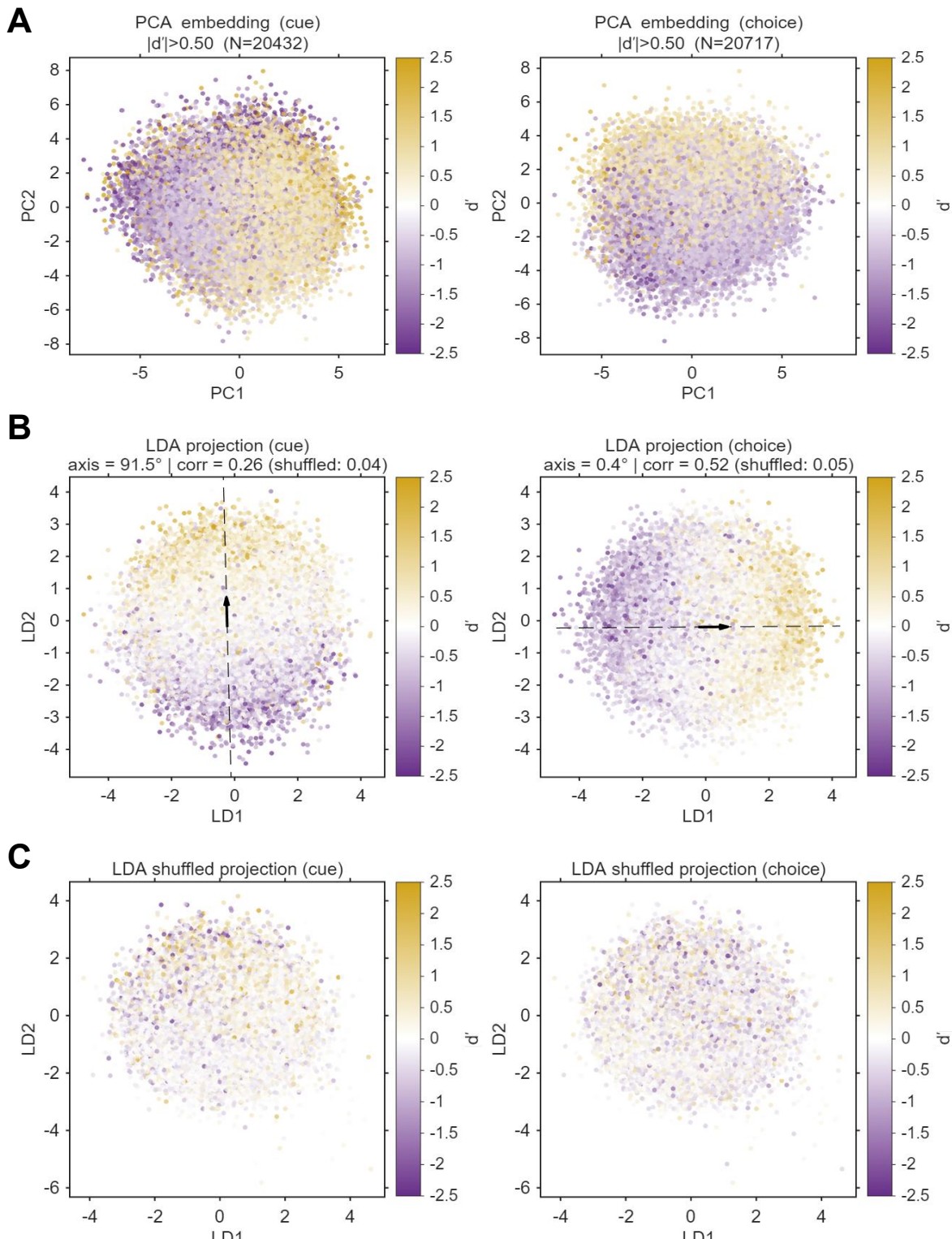

*Figure 8. Linear analysis of held-out dataset for CalM framework.* (A) PCA visualization for strong cue- and choice- tuning neurons. (B) LDA analysis shows similar orthogonal gradient structure. (C) Shuffle analysis for LDA.

