# OpenReview forum: "CalM: A Self-Supervised Foundation Model for Population Dynamics in Calcium Imaging Data"
_ICML.cc/2026/Conference — ICML 2026 regular_

### Official Review · Reviewer_xYgJ · 2026-02-19

**Soundness:** 3
**Presentation:** 3
**Significance:** 2
**Originality:** 2
**Overall Recommendation:** 5
**Confidence:** 4

**Summary:**

The manuscript proposes CalM, a self-supervised model for calcium-imaging population data composed of a Neural Quantizer, a VQ-VAE-style tokenizer that maps single-neuron traces into a shared set of discrete tokens, and a Dual-Axis Transformer that performs autoregressive modeling factorized across the neuron axis and the time axis. The authors demonstrate that CalM outperforms several baseline methods in neural future forecasting tasks and behavior decoding tasks.

**Compliance With Llm Reviewing Policy:**

Affirmed.

**Final Justification:**

My issues are resolved.

**Key Questions For Authors:**

Please see Strengths and Weaknesses.

**Limitations:**

Yes.

**Strengths And Weaknesses:**

Strengths
- Perturbing the teacher-forcing prediction strategy via scheduled sampling is a plausible and well-motivated idea. The authors provide the necessary ablations to demonstrate its impact on future forecasting performance.
- The manuscript is well written and easy to follow.
- The manuscript includes extensive comparisons across multiple baselines and downstream tasks, which helps contextualize the proposed method.

Weaknesses:
- The authors refer to “neural embeddings” and “session embeddings” in several places (e.g., Sections 5.4, 6.4, and Appendix E), but these are not described in the Methods section. The terminology is therefore unclear. Do these correspond to neuron/session identity embeddings similar to those used in NDT2 or POYO? If not, how exactly are they computed and incorporated into the model?
- No statistical significance tests are reported throughout the manuscript. In addition, the sample sizes used to compute the reported averages are not specified for any of the results. Reporting confidence intervals, standard errors, or statistical tests would strengthen the empirical claims.
- No codebase is provided, nor is there any mention of a future code release. Public availability of the code is important to support reproducible scientific research.
- The experiments are conducted exclusively on mouse data, whereas POCO was trained and evaluated on data from multiple species (including zebrafish and mice). Is there a specific reason for limiting the analysis to mouse data? If not, evaluating the model on additional datasets would significantly strengthen the generalization claims. For example, Allen Institute calcium imaging datasets could provide a large-scale multi-session benchmark.
- Training details such as the number of epochs, early stopping criteria, and hyperparameter tuning procedures are not provided for baseline methods such as POCO and POYO+. These factors can substantially influence performance and are important for ensuring fair comparisons.
- Although several ablations are included in the Appendix, they are barely discussed in the main text (except for L314). A dedicated paragraph summarizing the ablation findings would help readers better understand which components of the model are crucial for the observed performance gains.
- Can the authors comment on the necessity of training a VQ tokenizer prior to model training? Have they experimented with training the tokenizer end-to-end with the backbone, and performing predictions directly in the continuous observation space instead of the discrete token space? Clarifying this design choice would strengthen the methodological justification.

---

> ### Author Rebuttal · Authors · 2026-03-31
>
> We thank the reviewer for the helpful suggestions.
>
> 1, We apologize for the lack of clarity. CalM uses learnable neuron and session embeddings as adapters. Tokenized trace of each neuron is combined with specific neuron and corresponding session embedding before entering DAT. For unseen sessions, we add new session and neuron embeddings and optimize only them with the backbone frozen.
>
> 2, We report sample sizes, means, and standard deviations for all results (held-in: 189 sessions; held-out: 97; single-session: 3). Wilcoxon tests show that CalM significantly outperforms POYO+ on decoding and POCO on held-in forecasting, while the held-out forecasting difference is not statistically significant.
>
> VQ reconstruction:
> |Metric (Corr)|Held-in|Held-out|
> |---|---|---|
> |Train|0.9479$\pm$0.0142|0.9486$\pm$0.0147|
> |Eval|0.9473$\pm$0.0143|0.9479$\pm$0.0149|
> |Test|0.9473$\pm$0.0142|0.9482$\pm$0.0147|
>
> Forecasting:
> |Metric (Corr)|Real|Simulation|
> |---|---|---|
> |iTransformer|0.0822$\pm$0.0383|0.1941$\pm$0.0996|
> |PatchTST|0.1918$\pm$0.0526|0.2563$\pm$0.1501|
> |TCN|0.1854$\pm$0.0613|0.4077$\pm$0.1564|
> |POCO|0.2901$\pm$0.0783|0.4688$\pm$0.1365|
> |Ours|0.4025$\pm$0.0858|0.5207$\pm$0.0692|
>
> |Metric (Corr)|Held-in|Held-out|
> |---|---|---|
> |POCO|0.3668$\pm$0.0605|0.3322$\pm$0.0511|
> |Ours|0.3799$\pm$0.0978|0.3383$\pm$0.0860|
> |diff|+0.0131|+0.0061|
> |p-value|7.82e-04|0.3684|
> |significance|***|n.s.|
>
> Decoding:
>
> |Metric (R²)|Pitch|Roll|Yaw|
> |---|---|---|---|
> |GLM|0.6248$\pm$0.0864|0.5779$\pm$0.1190|0.5764$\pm$0.0650|
> |RRR|0.6220$\pm$0.0912|0.5787$\pm$0.1195|0.5764$\pm$0.0649|
> |TCN|0.4930$\pm$0.1783|0.4188$\pm$0.1956|0.3097$\pm$0.2510|
> |POYO+|0.6077$\pm$0.1664|0.6066$\pm$0.0719|0.6711$\pm$0.0725|
> |Ours-Linear|0.7017$\pm$0.1017|0.6595$\pm$0.0875|0.5745$\pm$0.1391|
> |Ours-Nonlinear|0.7715$\pm$0.0687|0.7314$\pm$0.1003|0.6493$\pm$0.0755|
>
> |Metric (R²)|Held-in pitch|Held-in roll|Held-in yaw|Held-out pitch|Held-out roll|Held-out yaw|
> |---|---|---|---|---|---|---|
> |POYO+|0.7227$\pm$0.0840|0.6751$\pm$0.1132|0.5853$\pm$0.2321|0.7073$\pm$0.0611|0.6446$\pm$0.1299|0.5430$\pm$0.2240|
> |Ours|0.7873$\pm$0.0737|0.7540$\pm$0.0883|0.7076$\pm$0.1691|0.7309$\pm$0.0785|0.6980$\pm$0.1200|0.6022$\pm$0.2172|
> |diff|+0.0647|+0.0790|+0.1223|+0.0236|+0.0534|+0.0592|
> |p-value|2.09e-32|1.47e-32|7.37e-30|2.59e-06|1.84e-10|2.14e-07|
> |significance|***|***|***|***|***|***|
>
> 3, We have provided the anonymous GitHub repository, including the training/evaluation code and the plotting scripts used to generate the quantitative results. Please refer to https://anonymous.4open.science/r/projectcode2026.
>
> 4, We chose the main dataset because, it provides both large total scale and substantial per-session neuron counts, making it suitable for validating large-scale multi-session modeling. To further address your concern, we additionally trained/evaluated CalM on an extra mouse dataset and a *C. elegans* dataset with spontaneous recodings. CalM still shows comparable results in forecasting tasks and performs better in decoding tasks under multi-session setup. For detailed information and results, please refer to reviewer **bucZ**, point 2 due to the word limit.
>
> 5, We train for 200 epochs, select the best checkpoint by validation performance, and confirm convergence by no improvement over the last 20 epochs. For POCO, we perform a broad hyperparameter search on the full held-in training set and select the best setting by validation metric. Due to its hyperparameter stability, we use a single configuration across all tasks, adjusting only learning rate and weight decay. For POYO+, we tune hyperparameters on a 9-session subset using validation MSE, then apply them to the full 189-session pre-training set with adjusted learning rate. For single-session decoding, we grid search model size, latent step, number of latents, and dropout ratio, and select by validation MSE. Final settings are listed in Tables 3-4.
>
> 6, Ablations show that for NQ, autoregressive regularization is the main driver of downstream forecasting gains, likely by encouraging temporally predictive tokens. Entropy and orthogonality regularization further help by preventing codebook collapse. For DAT, scheduled sampling is the most important component, as it mitigates exposure bias in long-horizon rollout, while neighborhood replacement provides a smaller gain by improving robustness to quantization errors.
>
> 7, We use two-stage training because calcium traces contain recurring transient patterns that are naturally compressible into a shared discrete vocabulary, enabling more stable and efficient large-scale autoregressive pretraining. End-to-end training is impractical due to non-differentiable argmax operations in discrete token selection during both encoding and decoding. As a proof of concept, we do train a comparable continuous model without VQ tokenizer (see response to Reviewer bucZ, under “key questions for the authors”), and CalM performs substantially better.
>
> We thank the reviewer and will revise the article.

---

> > ### Author Rebuttal · Reviewer_xYgJ · 2026-04-05
> >
> > I thank the authors for their rebuttal. My issues are resolved and I increased my score to 5.

---

> > > ### Author Response · Authors · 2026-04-05
> > >
> > > We thank the reviewer for the careful review and are glad that our rebuttal resolves your concerns.

---

### Official Review · Reviewer_bucZ · 2026-03-09

**Soundness:** 3
**Presentation:** 4
**Significance:** 3
**Originality:** 2
**Overall Recommendation:** 5
**Confidence:** 4

**Summary:**

This paper introduces a pre-training framework for calcium image traces that first trains a VQ-VAE to ensure the subsequent autoregressive model can be trained on discretized embeddings, instead of the raw time series. Tokenizing the timeseries in this way can help reduce noise, and makes the autoregressive task more similar to language-based modeling frameworks, which have shown to benefit greatly from autoregressive pre-training.

**Compliance With Llm Reviewing Policy:**

Affirmed.

**Final Justification:**

The authors were able to address most of the points that I raised in my initial review. Since my review was already positive with regards to acceptance, my final score stays the same, and I recommend acceptance of this work.

**Key Questions For Authors:**

Have the authors compared the performance of their model with and without discretization?

**Limitations:**

Yes

**Strengths And Weaknesses:**

**Strengths** \
The authors’ paper looks great, and is easy to read. The work introduces some interesting improvements over previous work, and compare against important baselines. Moreover, the authors show that their design choices clearly improve performance, and verify their model’s performance across a variety of tasks. The fact that their method outperforms a strong supervised baseline like POYO+, even with a linear head for certain tasks, highlights the strength of their pre-training method.

**Weaknesses** \
In Section 5.1 it sounds like the authors determine the best NQ model based on held-out data. If this is the case, the authors should re-run their experiments without selecting the best NQ model based on held-out data to avoid any potential leakage.

The authors should include another experimental dataset to further verify their results. Although the results on the current dataset are good, I believe it is important to show that the method generalizes to other datasets as well.

The authors should vary the weights in the loss function for their ablation study to better understand how weighing different aspects of the proposed loss functions interact or affect performance.

Although the low-dimensional dynamics results are interesting in Figure 5C, the assumption that PCA is a good ground-truth for the data’s intrinsic low-dimensional manifold is counterintuitive. Comparing the predicted and true dynamics makes sense, but I believe forecasting performance is already captured by the results in Figure 3B,C. The assumption that the PCA components span the ground-truth manifold runs counter to the fact that the authors use non-linear spatiotemporal methods to do the forecasting in the first place.

---

> ### Author Rebuttal · Authors · 2026-03-31
>
> We thank the reviewer for the helpful suggestions.
>
> Regarding the weaknesses:
>
> 1, We apologize for the lack of clarity. **For multi-session setting, the best NQ was chosen solely with held-in validation sets.** Specifically, NQ was trained on the held-in training sets, and the checkpoint with the best average performance on the held-in validation sets was selected, which was then evaluated on the four target splits (held-in test, held-out train, held-out validation, and held-out test). Therefore, held-out data was not used for model selection, and there is no data leakage.
>
> 2, To address this concern, during rebuttal time,we additionally trained/evaluated CalM on a mouse dataset and a *C. elegans* dataset with spontaneous recording on multi-session tasks.
>
> For the mouse dataset from Sun et. al. (2025), which involves a cue–delay–choice navigation task in which mice must use a transient visual cue to infer trial type and select the correct future reward location, we select a subset of 10 Hz sampling rates, which contains 13 subjects and a total of 160,182 neurons across 39 sessions (for held in, 10 subjects/30 sessions/128,421 neurons; for held out, 3 subjects/9 sessions/31,761 neurons). We pick up the first 64 time points of each trial for forecasting and decoding.
>
> For the *C. elegans* dataset from Atanas et al. (2023), which consists of brain-wide calcium imaging of spontaneous recording from freely moving animals. In the main analysis set, 14 animals exploring sparse food for 16 minutes were recorded at approximately 1.7 Hz, with an average of 143±12 head neurons per session. In our processed subset, we use 80 original HDF5 sessions after excluding scrambled controls, and split them into 60 held-in and 20 held-out sessions. The sessions are segmented into non-overlapping windows of 80 time points as pseudo-trials.
>
> For the mouse dataset:
>
> Forecasting:
>
> |Metric (correlation)|Held in|Held out|
> |---|---|---|
> |POCO|0.2781$\pm$0.0343|0.2568$\pm$0.0289|
> |CalM|0.2783$\pm$0.0562|0.2592$\pm$0.0364|
>
> Behavior decoding on position:
>
> |Metric (R2)|Held in|Held out|
> |---|---|---|
> |POYO+|0.8810$\pm$0.0745|0.8870$\pm$0.0649|
> |CalM|0.9706$\pm$0.0241|0.9581$\pm$0.0376|
>
> For the *C. elegans* dataset:
>
> Forecasting:
>
> |Metric (correlation)|Held in|Held out|
> |---|---|---|
> |POCO|0.1935$\pm$0.0670|0.1969$\pm$0.0468|
> |CalM|0.1841$\pm$0.0898|0.1601$\pm$0.0757|
>
> Behavior decoding on velocity:
>
> |Metric (R2)|Held in|Held out|
> |---|---|---|
> |POYO+|0.6922$\pm$0.2321|0.7394$\pm$0.0950|
> |CalM|0.7065$\pm$0.2658|0.7443$\pm$0.1010|
>
> Behavior decoding on head angle:
>
> |Metric (R2)|Held in|Held out|
> |---|---|---|
> |POYO+|0.4597$\pm$0.1469|0.4953$\pm$0.1145|
> |CalM|0.5149$\pm$0.1378|0.5606$\pm$0.0807|
>
> CalM achieves comparable performances on forecasting and performs better on decoding tasks.
>
> 3, Additional ablations show that scheduled sampling (SS) is critical for forecasting: weak scheduled sampling is sufficient, but removing it causes a clear drop, confirming its role in mitigating exposure bias. Neighborhood replacement (NR) yields a smaller but consistent gain, mainly by improving robustness to codebook vectors with small distances from NQ.
>
> Ablation:
> |SS prob|0.6|0.4|0.2|0.0|
> |---|---|---|---|---|
> |Metric (correlation)|0.3058±0.1098|0.3052±0.1076|0.2872±0.1084|0.2079±0.1027|
>
> |NR prob|0.10|0.06|0.03|0.00|
> |---|---|---|---|---|
> |Metric (correlation)|0.3058±0.1098|0.3057±0.1084|0.2986±0.1121|0.2870±0.1160|
>
> 4, Our goal in this analysis is not to assume that PCA provides the ground-truth intrinsic manifold of the neural activity. Instead, we use PCA as a simple tool to examine whether the model, despite not perfectly forecasting neural activity, still captures low-dimensional dynamics shared across the neural population with original traces. While learning neural data with high spatiotemporal complexity requires nonlinear methods, linear PCA can nevertheless provide a useful approximation for estimating and comparing dominant low-dimensional dynamics.
>
> Regarding the key question on discretization v.s. continuous modeling, we additionally constructed a continuous variant and compared it with the discretized version in the single-session setting. The continuous model has exactly the same DAT backbone, parameters and training setting with CalM, but accepts the calcium trial traces as input and is trained autoregressively with MSE loss. In forecasting case, we use the same setup as in Figure 3C. CalM performs much better than the continuous counterpart, highlighting the benefit of discretization.
>
> |Metric|Continuous model|CalM (discrete)|
> |---|---|---|
> |Correlation|0.2624$\pm$0.0519|0.4025$\pm$0.0858|
>
> Reference:
>
> Sun et. al. (2025) Learning produces an orthogonalized state machine in the hippocampus
>
> Atanas et al. (2023) Brain-wide representations of behavior spanning multiple timescales
>
> We thank the reviewer again and will revise the manuscript accordingly.

---

> > ### Author Rebuttal · Reviewer_bucZ · 2026-04-03
> >
> > Thank you for the rebuttal.
> > I will keep my score, since it already reflects an acceptance.
> > I would like to know though: Do the authors know why their model underperform on forecasting for the C. Elegans dataset?

---

> > > ### Author Response · Authors · 2026-04-05
> > >
> > > We thank the reviewer for acknowledging our work.
> > >
> > > A plausible explanation for the slightly weaker forecasting results on *C. elegans* is that these data are spontaneous recordings segmented into pseudo-trials, whereas CalM is more naturally suited to trial-aligned calcium dynamics.
> > >
> > > More generally, we find forecasting to be a challenging task in our experiments, and relative performance can vary across datasets and recording regimes, so we think this modest difference is not too surprising. Meanwhile, CalM remains consistently stronger on the downstream decoding tasks after pretraining.
> > >
> > > We will clarify this discussion in the revision. Thank you again for your thoughtful questions and careful review.

---

### Official Review · Reviewer_UMP5 · 2026-03-13

**Soundness:** 4
**Presentation:** 4
**Significance:** 4
**Originality:** 3
**Overall Recommendation:** 4
**Confidence:** 4

**Summary:**

Summary:
The paper describes an auto-regressive model of single neuron recordings in mice using calcium imaging. The method uses a VQ-VAE tokenization approach of calcium traces. Then it uses an auto-regressive transformer architecture to learn a self-supervised representation of the calcium data. An auxiliary task with behavior decoding is also used. The method is tested on held out sessions and animals from the Tseng et al. 2022 dataset from the Harvey lab. The authors report that than a non- linear attention strategy is improving the performance.

During evaluation only the session adapters are trained on new sessions. The model is evaluated by fine tuning 3 held out sessions, and tested on sequence forecasting conditioned on 40 time steps from real data.  The model is also evaluated on behavior decoding and similarity of PCA sub spaces.

**Compliance With Llm Reviewing Policy:**

Affirmed.

**Key Questions For Authors:**

Question to authors:
Please address the weaknesses that I mentioned above.

I suspect that  that the model shows at least some level of generalization during pre-training to the new sessions, and that the model can explain clearly what are the GLM and TCN baselines as discussed above. Assuming the authors will be capable to report some results in that direction I am inclined to accept the paper. I might reduce my grade if the authors fail to confirm this, or fail to explain why the model does not gain anything from the pre-training. I might also increase my grade, if the authors highlight an important point of the paper that I overlooked.

**Limitations:**

yes

**Strengths And Weaknesses:**

Strengths:

1) The model is very well defined. The model specifications is every clear and very reasonable in the light of the comparable models that have working in other fields like audio processing, EEG, fMRI processing etc...  To my knowledge, this is the first transformer model of this type for calcium imaging. I am convinced that a model of this type has some chances to become a good reference towards foundation models for calcium imaging data (it may need for effort like training on more datasets though, on the long run only time will tell whether the community agrees).

2) The model choices and evaluations are consistent with the existing literature. The POCO baseline is reasonable.

Weaknesses:
1) I miss clear evaluation of the model before and after pre-training. Considering the 3 sessions of held out sessions that is using for testing it seems that everything was planned so the quality of the pre-trained could be evaluated. With N epoch of fine-tuning on 3 sessions, what is the fit gain with fine-tuning. This seems like a very important data point that I could not find in the main figures.

2) I understand that the tokenization is done a single unit traces, and then that all the units from a sessions are projected in a session transformer independent backbone.  This choice seems very reasonable and it is in general very well explained. However, I could not fully understand where are the session specific adapters and how each tokenized channel is then combined and merged into the main transformer backbone. I would appreciate if the authors would clarify this or explain where I can find the information.

3) Regarding baseline I was surprising that a simple GLM was so good. For instance how can it be better than a TCN? Since GLM is basically an auto-regressive conv1d, I assume that a good TCN should beat the GLN baseline? It is also not clear to me how these models can be used with a pre-trained core and fine tuned on 3 sessions. So I wonder if some of these models are not pre-trained and should therefore be flagged as such.

4) I was surprised to see POYO+ as a baseline in Figure 4. I believe POYO was initially a model for extra cellular spike sorted spike trains. How is that directly comparable to a model of calcium imaging data? Is there a reasonable explanation why POYO+ is only code for a single task of decoding?

---

> ### Author Rebuttal · Authors · 2026-03-31
>
> We thank the reviewer for the helpful suggestions.
>
> Regarding the weaknesses:
>
> 1, We apologize for the lack of clarity. In Fig. 3C, all models are trained independently within each single session, so no fine-tuning is involved there. This experiment was intended to test whether CalM can already work well in the single-session setting, and we report the aggregated results across independently trained sessions. In contrast, Fig. 3D corresponds to the multi-session transfer setting, where the model is pretrained on the held-in dataset (189 sessions) and then adapted to the held-out sessions (97 sessions). For decoding, we replace the forecasting head with a task-specific head and fine-tune only that head. Therefore, in our paper, “fine-tuning” refers to lightweight adaptation to new animals/sessions/neurons or training a task-specific head, rather than end-to-end re-optimization of the pretrained backbone to maximize held-out performance.
>
> 2, We agree that this part should be explained more clearly in the paper. For each session and its corresponding neurons, CalM uses unique session embeddings and neuron embeddings as session-/neuron-specific adapters. These embeddings are learned jointly with the backbone during pretraining. When new sessions or neurons appear at adaptation time, we introduce new corresponding session/neuron embeddings, freeze the pretrained backbone, and optimize only these newly added embeddings. CalM performs well under this parameter-efficient fine-tuning (PEFT)-style adaptation paradigm, and we will revise both the text and figures to make this pipeline clearer.
>
> 3, We agree that the relatively strong GLM baseline and the comparison with TCN deserve clearer explanation. We adopt Ridge regression as our GLM baseline, which admits a closed-form solution that guarantees convergence to the global optimum. Although GLM has lower modeling capacity than TCN and can be regarded as a special case of Conv1D, on the small single-session datasets used in our evaluation, this optimization stability gives Ridge a practical advantage. In contrast, TCN relies on gradient-based training and is therefore more prone to suboptimal convergence on such limited data.
>
> In Fig. 4B, GLM and TCN are trained within a session in single-session experiments as described in 1, so there is no pretraining here. Testing results are reported from the average performances of the testing trials in each single-session dataset. And these methods can not be applied to multi-session experiments due to the lack of ability to deal with unseen neurons.
>
> 4, POYO+ is a multi-session, multi-task supervised decoding model that extends POYO so a single model can be trained jointly on data from different sessions, brain areas, and cell types for calcium traces. As it was published on ICLR2025, please refer to the link for the paper: https://openreview.net/pdf?id=IuU0wcO0mo.  POYO+ is used in our paper only as a multi-session supervised decoding baseline, not as a directly equivalent foundation-model baseline for all tasks. We included it because, to the best of our knowledge, it is one of the strongest recent models for multi-session supervised decoding in calcium trace data settings and therefore provides a meaningful point of comparison in our multi-session decoding experiments.
>
> Despite the points for weaknesses we have answered, regarding your question on whether there is an important point that may have been overlooked, we would like to emphasize that CalM is designed as a unified framework in which the same pretrained backbone supports both generative forecasting and downstream decoding through lightweight embedding tuning. The main contribution is that a pretrained calcium model can generalize to more unseen sessions/animals with the backbone frozen and only minimal session/neuron adaptation or task-head replacement. At the same time, we additionally train/evaluate the model on extra mouse and *C. elegans* datasets and show the broad effectiveness of CalM, which is elaborated in details in the answer to reviewer bucZ, point 2.
>
> We thank the reviewer again for the helpful comments and will revise the manuscript accordingly.

---

> > ### Author Rebuttal · Reviewer_UMP5 · 2026-04-02
> >
> > Thank you for the response.
> >
> > I think the clarification from the authors are useful and clear for the points 1,2, and 4. To my understanding, my grade was appropriate.
> >
> > I do think that my point 3 might raise in important reg flag. I find it surprising that Ridge regression with 1d convolution becomes such a great competitor. It may highlight that other baselines like TCN are problematic.

---

> > > ### Author Response · Authors · 2026-04-08
> > >
> > > Dear Reviewer,
> > >
> > > We are glad that our clarifications on points 1, 2, and 4 are helpful. Regarding point 3, we agree that the unexpected GLM-vs-TCN comparison makes this result worth examining more carefully, and we thank you for raising this concern.
> > >
> > > To better understand this phenomenon, we revisit the single-session decoding baselines described in the appendix and provide the corresponding implementation in the baseline section of our anonymous repository: https://anonymous.4open.science/r/projectcode2026/baseline/run_baselines_glm_tcn_final.py
> > >
> > > To assess the effect of data variability, we evaluated the methods both on the original 3-session single-session split used in the paper and on three additional 3-session groups. Meanwhile, to better understand how model design affects TCN performance, we design a TCN variant with residual connections and optimize it by grid search.
> > >
> > > The results are shown below:
> > >
> > > Data in the paper setting (3 sessions):
> > >
> > > | Methods / decoding variables | **Pitch** | **Roll** | **Yaw** |
> > > | --- | --- | --- | --- |
> > > | GLM | 0.6248±0.0706 | 0.5778±0.0971 | 0.5763±0.0530 |
> > > | TCN | 0.4930±0.1783 | 0.4188±0.1956 | 0.3097±0.2510 |
> > > | TCN with residual connection | 0.6217±0.0948 | 0.6007±0.1145 | 0.6076±0.0762 |
> > > | CalM | 0.7715±0.0687 | 0.7314±0.1003 | 0.6493±0.0755 |
> > >
> > > Data group 1 (3 sessions):
> > >
> > > | Methods / decoding variables | Pitch | Roll | Yaw |
> > > | --- | --- | --- | --- |
> > > | GLM | 0.4113±0.1798 | 0.3789±0.1566 | 0.5734±0.0863 |
> > > | TCN | 0.4772±0.1260 | 0.4216±0.1527 | 0.5462±0.1038 |
> > > | TCN with residual connection | 0.5861±0.1290 | 0.6081±0.1065 | 0.7689±0.0502 |
> > > | CalM | 0.8004±0.0487 | 0.8011±0.0457 | 0.7986±0.0684 |
> > >
> > > Data group 2 (3 sessions):
> > >
> > > | Methods / decoding variables | Pitch | Roll | Yaw |
> > > | --- | --- | --- | --- |
> > > | GLM | 0.6017±0.0742       | 0.3568±0.1374 | 0.5384±0.0340 |
> > > | TCN | 0.6119±0.0140 | 0.4169±0.1361 | 0.5246±0.0327 |
> > > | TCN with residual connection | 0.6801±0.0387 | 0.5549±0.1178 | 0.7272±0.0458 |
> > > | CalM | 0.8039±0.0261 | 0.7169±0.0479 | 0.7286±0.0208 |
> > >
> > > Data group 3 (3 sessions):
> > >
> > > | Methods / decoding variables | Pitch | Roll | Yaw |
> > > | --- | --- | --- | --- |
> > > | GLM | 0.6843±0.0284       | 0.5539±0.1145 | 0.6667±0.1041 |
> > > | TCN | 0.6233±0.0623 | 0.5465±0.0734 | 0.5381±0.0812 |
> > > | TCN with residual connection | 0.7118±0.0331 | 0.6411±0.0560 | 0.7754±0.0256  |
> > > | CalM | 0.8412±0.0205 | 0.7892±0.0278 | 0.7941±0.0311  |
> > >
> > > Based on these additional experiments, we conclude that, beyond the optimization-related explanation we discussed in the previous rebuttal, the performances of GLM and TCN are also sensitive to TCN design and session variability. In particular, with residual connections, the TCN becomes substantially stronger and outperforms the previous TCN and GLM. This may suggest that the original comparison should not be over-interpreted, and that the observed behavior partly reflects architectural choice and data variability.
> > >
> > > While the augmented TCN performs substantially better, it still underperforms CalM both in the experiments of the paper setting and the extra data, and the trends for each variable reported in the original paper for the single-session setting are preserved on the extra data, so the conclusion does not change.
> > >
> > > Finally, we would like to note that our main contribution lies in the multi-session setting. Meanwhile, we made much effort to grid search the hyperparameters for the best performance of baselines in our experiments to make the comparisons as fair as possible.
> > >
> > > Thank you again sincerely for helping us examine this key question. We will revise the manuscript to clarify the difference, and update the baseline discussion to present the comparison more clearly.

---

### Official Review · Reviewer_U8FG · 2026-03-14

**Soundness:** 3
**Presentation:** 3
**Significance:** 2
**Originality:** 2
**Overall Recommendation:** 4
**Confidence:** 4

**Summary:**

This paper proposes CaIM, a two-stage self-supervised foundation model trained only on neuronal calcium traces and adaptable to downstream tasks. It is comprised of a VQ-VAE tokenizer converting single-neuron traces into a shared discrete vocabulary, and dual-axis autoregressive transformer to model dependencies across neurons and across time. After pre-training, it can be applied to different problems by using a task-specific head.

CaIM is evaluated on simulated data and an open-source dataset of mice performing a navigation task, including multiple animals and sessions and hundreds of thousands of neurons in total. The benchmark includes two tasks: population dynamics forecasting --  on which CaIM achieves competitive performance -- and behavior decoding, on which CaIM achieves superior results against specialized models such as POYO+. Linear analysis also suggests that the learned representations capture realistic low-dimensional neuronal response structure.

**Compliance With Llm Reviewing Policy:**

Affirmed.

**Final Justification:**

The authors have clarified several of my questions and made the code available, increasing the Soundness aspect so I am happy to increase my score to a 4. I still believe this is an overly complex approach though, with somewhat weak evidence for generalization to other datasets.

**Key Questions For Authors:**

(i) Can the model trained on certain areas be expected to perform well when applied to other brain areas? Or would it require to be retrained from scratch? In what scenarios should one safely reuse the pretrained model? When should one retrain the NQ or the DAT backbone?

(ii) What are the normalization/standardization steps needed to preprocess the calcium traces? Would it work with different sampling rates? Including a quick-reference step-by-step guide for how to use this model in practice would greatly increase the contribution of this paper.

(iii) Could the authors provide the full code used to produce the results and figures (using something like Anonymous GitHub)? This would ensure reproducibility by other labs as well as confidence on the paper claims.

(iv) Why haven't the authors tested their model on other publicly available datasets of a similar kind: e.g., the calcium imaging data from mouse visual cortex available from the Allen Brain Observatory?

(v) Do the authors have an idea of how much data (# neurons, or recording length) is required to achieve the results reported? Providing as estimate of this would be important for other labs to consider using this model.

**Limitations:**

The only limitation that I found mentioned was "CalM relies on trial-aligned data, which may limit the broader usage." Other shortcomings were only indirectly mentioned and treated more as future directions. I believe this paper would benefit if the authors critically assessed its current limitations, ideally in a dedicated paragraph within the Discussion. As examples, they could discuss the necessity of trial structure; the complexity of the multi-stage pipeline; the amount of data required for fine-tuning/re-training the model to other recordings; and possibly how challenging its pretraining may be; and the extent to which CaIM is expected generalize beyond the datasets used in this study.

**Strengths And Weaknesses:**

__Strengths:__


The study uses a large-scale dataset that includes multiple animals and sessions.

The proposed tokenizer seems to work well, according to Figure 1: high quality reconstruction and generalization to held out data.

The ablation study performed helps confirm that the details of the architecture proposed are indeed relevant.

The model achieved comparable or better results against other SOTA methods (POCO, POYO+).

The paper also presents a linear analysis for interpretability of the model's learned representations.

__Weaknesses:__

Although the real-world dataset used for training and validation is large, it is a single dataset, with not very many animals, limited to 2 cortical layers, and consisting of a single task. This severely limits the extent to which the paper's conclusions can be expected to generalize to other scenarios.

The PCA analysis for interpretability could be more thorough: separating two expected functional groupings of neuronal responses (cue vs. choice) is a fairly basic result. Thus, it offers limited evidence that the model can actually capture any rich biological structure. For example, the authors could include more quantitative and fine-grained tests (using TDA analysis of the embeddings) to check if the learned representations are stable across sessions and/or animals.

Crucially, after reading the paper, I am left without clearly understanding how to apply this model to a new dataset (different brain areas/calcium dynamics; different species). For example: what should one do if the data has no labels, how much data is needed, and if it would be necessary to retrain the tokenizer.

---

> ### Author Rebuttal · Authors · 2026-03-31
>
> We thank the reviewer for the helpful suggestions.
>
> Regarding the weaknesses:
>
> We agree that, at submission time, our real-data evaluation was limited to one large calcium-imaging dataset. To address this concern, we additionally train/evaluate CalM on a mouse dataset and a *C. elegans* dataset and report the corresponding results in the revision.
>
> We also agree that the current PCA analysis for neural embedding is only a first-step interpretability probe. In Figure 5B, we provide further evidence that the neural embeddings form an approximately orthogonal gradient structure rather than simple clustering for neurons tuned to different task-related variable with LDA. This structure preserves in both held-in and held-out data. In our framework, the neuron embeddings exhibit high-dimensional representations of single-neuron functional identity. Consequently, they do not exhibit coherent topological structures (e.g., persistent H1) that would make standard topological data analysis particularly informative. More fine-grained analyses, including nonlinear analyses, are important future directions.
>
> If a new dataset has no labels, CalM can still be pretrained in a self-supervised autoregressive manner and adapted to new sessions; labels are only required for supervised downstream tasks such as behavior decoding.
>
> Regarding key questions for authors:
>
> 1, As a proof of concept during rebuttal, we retrain CalM on 6 sessions from V1 region and adapted it to 6 sessions from another brain region (RSC). During adaptation, for NQ, we do zero-shot inference; for DAT, we finetune the neural and session embeddings with the backbone frozen. We do bidirectional testing and CalM provides transferrable results.
>
> | pretrain region \ finetune region | V1 | RSC |
> | --- | --- | --- |
> | V1 | 0.4892$\pm$0.0281 | 0.4502$\pm$0.0176 |
> | RSC | 0.4587$\pm$0.0250 | 0.4832$\pm$0.0195 |
>
> More broadly, we expect pretrained NQ/DAT to be reusable when the new data are sufficiently similar in modality, temporal dynamics and animal tasks; if NQ reconstruction quality or downstream forecasting performance drops substantially, retraining or re-adaptation would be necessary in practice.
>
> 2, We preprocess the neural signals by first applying an exponential moving average for temporal smoothing of calcium traces. The smoothed traces are then segmented into trials, which are randomly split into training, validation, and test sets. Finally, we compute the per-neuron mean and standard deviation using only the training split and use these statistics to z-score normalize the neural data in all three splits.
>
> If the datasets contain captures with different sampling rates, we have not yet designed a fully flexible module for varying calcium sampling rates, and we agree this is an important next step to deal with diverse calcium datasets. Since NQ focuses more on calcium-trace shape than absolute frame count, one promising direction is to replace the first and last layers with adaptive convolutional layers that can accommodate different frame-rate window sizes while keeping the smallest time stride fixed.
>
> We have provided the step-by-step guide of training the model and dataset curation in the readme file in the anonymous github, please refer to https://anonymous.4open.science/r/projectcode2026.
>
> 3, We have provided the CalM code and the plotting code for results in the same link above.
>
> 4, To address this concern, we additionally test CalM on an extra mouse dataset and a *C. elegans* dataset with spontaneous recodings. CalM still shows comparable results in forecasting tasks and performs better in decoding tasks under multi-session setup. For detailed results, please refer to reviewer bucZ, point 2 due to the word limit.
>
> 5, In our current experiments, single-session data already supported NQ/DAT training in the single-session setting, corresponding to roughly 500–1500 neurons and around 400 trials per session, ~90 frames per trials (Appendix Table 1), in total around 36000 frames per session.  In simulation dataset including 200 neurons, 400 trials and 100 frames per trial, CalM also works. CalM would also be effective with more data, as shown in multi-session results.
>
> We thank the reviewer again for the helpful comments and will revise the manuscript accordingly.

---

> > ### Author Rebuttal · Reviewer_U8FG · 2026-04-02
> >
> > Thank you for the response.
> >
> > The clarification on estimates on # neurons, trials, frames provided are helpful and directly address the question.
> >
> > The additional experiments on mouse C. elegans, as well as the experiment comparing V1 and RSC are helpful and partially alleviate my concern on generalization. However, the evidence is still limited and somewhat anecdotal (e.g., small number of sessions, no systematic comparison across diverse datasets or modalities).
> >
> > The main concern about the interpretability analysis remains: there's limited evidence of any rich biological structure. The argument against TDA is reasonable, but additional quantitative analyses would be beneficial. This remains largely unresolved.
> >
> > Moreover, the author's rebuttal still leaves ambiguity about: how sensitive performance of the model is to the preprocessing choices, and concrete decision rules for retraining NQ vs. DAT.
> >
> > Finally, although the authors shared a repo with code, the readme is very far from a true quickstart guide (which would be essential given that over 100 files are provided). There are no clear instruction on how run their code and reproduce their main results from scratch. E.g., it doesn't seem to specify the software environment; there is no clear reproduction path from commands to paper results/figures.
> >
> > Overall, while this rebuttal improves on the clarity of a few points, my core concerns remain. Evidence for generalization and interpretability of the results remains limited; there is unclear practical guidance on how to apply the method to new, real-world datasets. In summary, this seems like a complex approach with a insufficient level of empirical validation and usability demonstrated so far.

---

> > > ### Author Response · Authors · 2026-04-08
> > >
> > > Thank you for the review.
> > >
> > > 1, Regarding generalization and retraining rule, we clarify that our paper did not make a broad claim across datasets or modalities. To address your concern, we additionally test cross-dataset transfer of the multi-session pretrained model from the Tseng dataset to the extra mouse dataset with 39 sessions (point 2, reviewer bucZ).
> > >
> > > For the tokenizer, NQ remains strong under transfer:
> > >
> > > |Corr|Train set|Val set|Test set|
> > > |-|-|-|-|
> > > |Retrain NQ|0.9119±0.0098|0.9123±0.0105|0.9122±0.0099|
> > > |Tseng NQ (inference only)|0.9046±0.0204|0.9051±0.0209|0.9050±0.0210|
> > >
> > > For downstream transfer, we compare 3 settings:
> > >
> > > |Setting|Forecast corr|Mean decoding R²|
> > > |-|-|-|
> > > |Retrain NQ & DAT|0.2764±0.0520|0.9710±0.0249|
> > > |Tseng NQ & Retrain DAT|0.2859±0.0568|0.9709±0.0314|
> > > |Tseng NQ & DAT(update embedding only)|0.2730±0.0586|0.9527±0.0622|
> > >
> > > These results provide evidence of cross-dataset generalization in this setting.
> > >
> > > More generally, we do not think a concrete, universally valid reuse/retraining rule can be specified in advance. Under domain shift, generalization to a new dataset depends on source-target relatedness and feature/task specificity (Ben-David et al.; Yosinski et al.). What we can provide is a practical heuristic: (1) reuse NQ by direct inference; if the target dataset shows clear mismatch in low-level signal characteristics/statistics, retrain NQ. (2) If NQ is reusable, reuse the pretrained DAT backbone with frozen weights and adapt only session/neuron embeddings. (3) If target-domain forecasting/decoding remains unsatisfactory after standard adaptation, further retrain DAT.
> > >
> > > 2, Regarding interpretability, we clarify that our paper did not claim that CalM captures “rich biological structure” in a broad mechanistic sense. Rather, our actual claim is that the learned representations capture low-dimensional dynamics and provide functional organization. We therefore believe the current concern partly evaluates the paper against a stronger interpretability target than what is claimed.
> > >
> > > We also do not think the LDA result is trivial. Rather than merely showing separability, it tests whether pooled neuron embeddings in a shared multi-session space exhibit structured task-related organization across sessions and animals. LDA reveals continuous tuning-strength gradients and approximately orthogonal cue/choice directions, suggesting graded functional identity and relatively disentangled cue/choice encoding. Traditional single-session analyses may detect within-session selectivity, but do not naturally provide such a shared space without extra feature engineering or alignment assumptions, especially when neuron identities and recording lengths differ across sessions.
> > >
> > > Richer biological analyses are certainly valuable, but we view them as future work rather than a prerequisite for the narrower claim made here.
> > >
> > > 3, Regarding the extra question on preprocessing sensitivity, we use the same preprocessing pipeline in all experiments. To address this concern, we additionally vary the EMA smoothing coefficient in the single-session setting. Preprocessing mainly affects reconstruction/forecasting correlation, while decoding remains stable. We also observe the same qualitative trend: from α = 0.15 to 1.00, forecasting correlation performance drops by 34.7% for CalM and 39.3% for POCO, suggesting that this sensitivity is not unique to CalM, while CalM still performs better.
> > >
> > > |Alpha|NQ Recon|DAT Forecast|POCO Forecast|Mean decoding R²|
> > > |- |-|-|-|-|
> > > |0.15 (default) |0.9561±0.0114|0.4600±0.0767|0.3328±0.0706|0.7499±0.0138|
> > > |0.60|0.8984±0.0100|0.3603±0.0727|0.2470±0.0501|0.7746±0.0192|
> > > |1.00 (w/o smoothing)|0.8784±0.0213|0.3003±0.0769|0.2020±0.0450|0.7796±0.0258|
> > >
> > > 4, Regarding code, In addition to the detailed instructions already provided, we added clearer environment instructions, a practical quickstart, explicit commands and clearer mappings from scripts to the main results/figures in the anonymous repository.
> > >
> > > Finally, regarding the reviewer's summary of CalM, we respectfully defend our conference submission:
> > >
> > > On complexity, while CalM is a two-stage framework, such modular designs are common in modern deep learning and compneuro, and the decomposition is directly motivated by the structure of calcium traces and scalable autoregressive pretraining.
> > >
> > > On validation, beyond the original large-scale benchmark, we additionally provide cross-dataset transfer evidence during rebuttal, while keeping baseline comparisons as fair as possible through careful tuning under the same evaluation protocol.
> > >
> > > On usability, we provide code, plotting scripts, and clearer practical instructions.
> > >
> > > We acknowledge the reviewer’s remaining concerns, but we hope the paper can be assessed with a more balanced summary of both its limitations and demonstrated strengths.
> > >
> > > References:
> > >
> > > Ben-David et al., 2010, *A Theory of Learning from Different Domains*
> > >
> > > Yosinski et al., 2014, *How transferable are features in deep neural networks?*

---

### Decision · Program_Chairs · 2026-04-30

**Decision:**

Accept (regular)

**Comment:**

This work introduces a tokenization and transformer model (termed CaIM) that aims to be a foundation model for calcium imaging. As calcium imaging is a primary mode of neural recording, and foundation models are useful in some settings, the work here would be potentially impactful to those who want to use a transformer model but have limited data to train such a model on. The initial review for this work noted a number of potential concerns, including what I found to be the main shared points of 1) reproducibility, 2) demonstration that the model was truly "foundational" and could be applied to other datasets, 3) questions on the performance of the included baselines. The authors responded with additional datasets (2 more), additional analyses, and clarifications that seemed to answer all the reviewers main concerns. I therefore recommend this work for acceptance.